# Facet sensitivity of iron carbides in Fischer-Tropsch synthesis

Wenlong Wu [1,2,3,5], Jiahua Luo[2,5], Jiankang Zhao[2], Menglin Wang[2], Lei Luo [2], Sunpei Hu[2], Bingxuan He[2], Chao Ma [4], Hongliang Li [2,3] ✉ & Jie Zeng [1,2] ✉

Fischer-Tropsch synthesis (FTS) is a structure-sensitive reaction of which performance is strongly related to the active phase, particle size, and exposed facets. Compared with the full-pledged investigation on the active phase and particle size, the facet effect has been limited to theoretical studies or single-crystal surfaces, lacking experimental reports of practical catalysts, especially for Fe-based catalysts. Herein, we demonstrate the facet sensitivity of iron carbides in FTS. As the prerequisite, {202} and {112} facets of χ-Fe$_5$C$_2$ are fabricated as the outer shell through the conformal reconstruction of Fe$_3$O$_4$ nanocubes and octahedra, as the inner cores, respectively. During FTS, the activity and stability are highly sensitive to the exposed facet of iron carbides, whereas the facet sensitivity is not prominent for the chain growth. According to mechanistic studies, {202} χ-Fe$_5$C$_2$ surfaces follow hydrogen-assisted CO dissociation which lowers the activation energy compared with the direct CO dissociation over {112} surfaces, affording the high FTS activity.

Fischer-Tropsch synthesis (FTS) is a structure-sensitive reaction for the sustainable production of synthetic fuels and building-block chemicals from syngas[1–6]. The catalytic performance is strongly related to the active phase, particle size, and exposed facets of the active components such as iron, cobalt, and ruthenium[7–13]. Compared with Co and Ru, Fe-based catalysts have superior properties including resistance to the formation of methane, low cost, high adaptability to broad H$_2$/CO ratios, and versatility to various useful products[14–16]. For Fe-based catalysts, pure-phase iron carbides were synthesized by using Fe(CO)$_5$ reagent, which was explored by means of in-situ characterizations[17,18]. The Fe$_3$O$_4$@χ-Fe$_5$C$_2$ core-shell catalysts were constructed by pyrolyzing iron-containing metal-organic frameworks, whereas the obtained nanoparticles were irregular spherical particles[19]. The transformation of reduced iron phases to iron carbides promoted the formation of hydrocarbon species in FTS[20]. The effects of the active phase and particle size have been extensively studied[19,21–26]. These effects are generally entangled with the contributions of surface terrace, corner, edge, and step-edge sites, where differences in coordination numbers and surface topology may lead to substantial differences in intrinsic performance[27]. Up to date, the investigation of the facet effect has been limited to theoretical calculations or single-crystal surfaces. For instance, density functional theory (DFT) calculations of CO activation on χ-Fe$_5$C$_2$ surfaces indicated that the terraced (510) surface inclined to directly dissociate CO molecules, whereas the stepped (010) and (001) surfaces preferred the hydrogen-assisted CO dissociation route[28]. In-situ scanning tunneling microscopy visualized on-surface ethylene polymerization on a carburized Fe(110) single-crystal surface[29]. However, there is no experimental report on the facet effect of practical Fe-based catalysts due to the complexity and dynamic structural evolution of iron carbides during FTS.

As the widely accepted active phase for FTS, χ-Fe$_5$C$_2$ has a base-centered monoclinic (bcm) structure with space group C2/c (a = 11.56 Å, b = 4.57 Å, c = 5.06 Å, and β = 97.74°)[30,31]. Owing to the low symmetry of the lattice structure, it remains as a grand challenge to

[1]School of Chemistry & Chemical Engineering, Anhui University of Technology, Ma'anshan, Anhui 243002, P. R. China. [2]Hefei National Research Center for Physical Sciences at the Microscale, Key Laboratory of Strongly-Coupled Quantum Matter Physics of Chinese Academy of Sciences, Key Laboratory of Surface and Interface Chemistry and Energy Catalysis of Anhui Higher Education Institutes, Department of Chemical Physics, University of Science and Technology of China, Hefei, Anhui 230026, P. R. China. [3]National Synchrotron Radiation Laboratory, University of Science and Technology of China, Hefei, Anhui 230026, P. R. China. [4]College of Materials Science and Engineering, Hunan University, Changsha 410082, P. R. China. [5]These authors contributed equally: Wenlong Wu, Jiahua Luo. ✉e-mail: lihl@ustc.edu.cn; zengj@ustc.edu.cn

synthesize χ-Fe₅C₂ nanocrystals with uniformly exposed surfaces. We proposed to use a highly symmetrical template as the core to support the χ-Fe₅C₂. Fe₃O₄ has a face-centered cubic (*fcc*) structure with a high symmetry. Herein, we reported the conformal reconstruction of well-defined Fe₃O₄ nanocrystals to generate χ-Fe₅C₂ with specifically exposed surfaces. The samples consisted of an inner core of Fe₃O₄ and an outer shell of χ-Fe₅C₂, denoted as Fe₃O₄@χ-Fe₅C₂ nanocrystals. We obtained Fe₃O₄@χ-Fe₅C₂ nanocrystals with surfaces terminated in {202} and {112} facets of χ-Fe₅C₂ shells through using cubic and octahedral Fe₃O₄ as the templates, respectively (Fe₃O₄@χ-Fe₅C₂ nanocubes and octahedra, respectively). We discovered that Fe₃O₄@χ-Fe₅C₂ nanocubes were more catalytically active and stable than the octahedral counterpart during FTS, whereas the facet sensitivity was not prominent for the chain growth. According to mechanistic studies, the high activity of {202} χ-Fe₅C₂ surfaces derived from the unique reaction path in which the hydrogen-assisted CO dissociation route

lowered the activation energy relative to the direct CO dissociation route over {112} surfaces.

## Results and discussion

### Synthesis and characterization of Fe₃O₄@χ-Fe₅C₂ nanocubes

To begin with, Fe₃O₄ nanocubes were prepared with an average size of $40.5 \pm 3.9$ nm and a purity of 95.3% (Supplementary Fig. 1). Fe₃O₄@χ-Fe₅C₂ core-shell nanocubes were synthesized via surface reconstruction of Fe₃O₄ under syngas atmosphere. Specifically, Fe₃O₄ nanocubes were reduced under 1 bar of H₂ with a gas-flow rate of 100 mL min⁻¹ at 270 °C for 10 h. This treatment ensured the removal of surface organic species (Supplementary Fig. 2). Afterwards, the obtained samples underwent surface reconstruction in a fixed-bed reactor under 20 bar of syngas (32 vol% H₂, 64 vol% CO, and 4 vol% Ar) with a space velocity of 2400 mL h⁻¹ g$_{cat}$⁻¹ at 270 °C for 20 h. Figure 1a shows the transmission electron microscopy (TEM) image of Fe₃O₄@χ-Fe₅C₂ nanocubes.

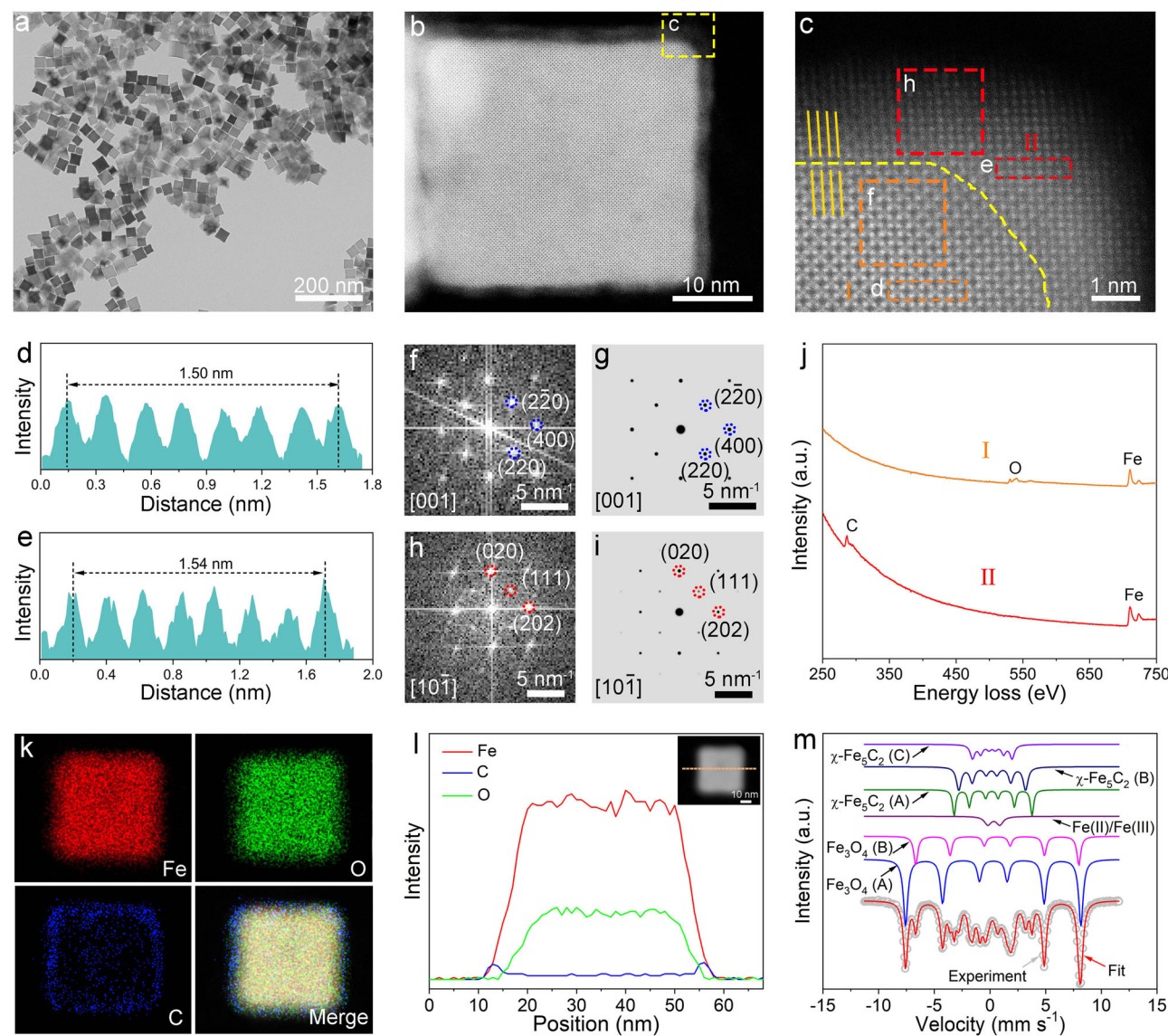

**Fig. 1 | Structural characterizations of Fe₃O₄@χ-Fe₅C₂ nanocubes. a** TEM image of Fe₃O₄@χ-Fe₅C₂ nanocubes. **b** HAADF-STEM image of an individual Fe₃O₄@χ-Fe₅C₂ nanocube. **c** Magnified HAADF-STEM image of the region marked by the corresponding box in panel (**b**). **d** Intensity profile recorded from the area indicated by the rectangular box in panel (**c**). **e** Intensity profile recorded from the area indicated by the rectangular box in panel (**c**). **f** FFT pattern from box **f** in panel (**c**).
**g** Simulated FFT pattern of Fe₃O₄ along the [001] direction. **h** FFT pattern from box **h** in panel (**c**). **i** Simulated FFT pattern of χ-Fe₅C₂ along the [10-1] direction. **j** EELS spectra of a Fe₃O₄@χ-Fe₅C₂ nanocube in panel (**c**). **k** Elemental mapping images of a Fe₃O₄@χ-Fe₅C₂ nanocube. **l** Line-scan profile recorded along the line of the inset HAADF image. **m** Mössbauer spectra of Fe₃O₄@χ-Fe₅C₂ nanocubes.

The average size of $Fe_3O_4$@χ-$Fe_5C_2$ nanocubes was 40.1 ± 3.8 nm with a purity of 90.0% (Supplementary Fig. 3). The $Fe_3O_4$@χ-$Fe_5C_2$ nanocubes exhibited a surface area of 20.4 $m^2 g^{-1}$ based on the Brunauer-Emmett-Teller (BET) method (Supplementary Fig. 4a). The high-angle annular dark-field scanning transmission electron microscopy (HAADF-STEM) image taken from one of the nanocubes revealed a periodic lattice extending across the entire surface (Fig. 1b). The magnified image clearly revealed lattice differences between the edge region and the central region, implying the formation of the core-shell structure (Fig. 1c). In the case of nanocrystals with a core-shell structure, a slight lattice mismatch ($f < ~5\%$) is required for the epitaxial surface layer to form over the inner core[32]. This epitaxial relationship allows for the maintenance of orientation between the growth layer and the substrate within the first few atomic layers. The spacing of $Fe_3O_4$(400) planes (0.21 nm) in the core was approximately equal to that of χ-$Fe_5C_2$(202) planes (0.22 nm) in the shell (Fig. 1d, e). The small lattice mismatch ($f$) of 4.65% calculated from Eq. (1) ensures the preservation of the epitaxial orientation relationship.

$$f = 2 \times |d_{shell} - d_{core}|/(d_{shell} + d_{core}) \tag{1}$$

In Eq. (1), $d_{shell}$ and $d_{core}$ refer to the lattice spacings of the shell and the core, respectively. The small lattice mismatch results in only one type of facet at the surface layer of $Fe_3O_4$@χ-$Fe_5C_2$ nanocubes. To identify the corresponding facets, we conducted the fast Fourier transform (FFT) analysis. According to the combination of the experimental and simulated FFT patterns, the region in the inner core was indexed as the (400) facet of *fcc* $Fe_3O_4$ along the [001] zone axis, while that in the outer shell corresponded to the (202) facet of *bcm* χ-$Fe_5C_2$ along the [10−1] zone axis (Fig. 1f−i). The HAADF-STEM images of different corners and edges in an individual nanoparticle indicated that the surfaces exposed a uniform χ-$Fe_5C_2$ shell with the {202} facets (Supplementary Fig. 5). The average thickness of the shell was determined as 2.0 nm, approximating ten atomic layers (Supplementary Fig. 5).

The core of iron oxides and the shell of iron carbides were further confirmed through the spatial elemental analysis. Specifically, the electron energy loss spectroscopy (EELS) image implied that the core region mainly comprised Fe and O elements while the shell region contained Fe and C elements (Fig. 1j). The scanning transmission electron microscopy-energy dispersive X-ray (STEM-EDX) analysis including the elemental mapping images and the line-scanning profile confirmed that O and C elements were mainly located in the core and the shell, respectively, while Fe was homogeneously distributed throughout the particle (Fig. 1k, l).

To quantify the contents of different phases in $Fe_3O_4$@χ-$Fe_5C_2$ nanocubes, we carried out Mössbauer and X-ray diffraction (XRD) characterizations. Based on the Mössbauer results, the sample was composed of 63.4 wt% of $Fe_3O_4$, 33.2 wt% of χ-$Fe_5C_2$, and 3.4 wt% of Fe(II)/Fe(III) species (Fig. 1m and Supplementary Table 1). Fe(II)/Fe(III) species indicated the presence of some poorly crystallized iron oxides. The quantitative analysis of XRD showed that χ-$Fe_5C_2$ occupied 29.8 wt% of the total mass, approaching that (33.2 wt%) obtained from the Mössbauer spectra (Supplementary Fig. 6a and Supplementary Table 2).

### Evolution from $Fe_3O_4$ nanocubes to $Fe_3O_4$@χ-$Fe_5C_2$ nanocubes

Figure 2a depicts the schematic of the evolution from $Fe_3O_4$ nanocubes to $Fe_3O_4$@χ-$Fe_5C_2$ nanocubes. The major steps were verified by HAADF-STEM images and XRD patterns. Specifically, after reduction, the initial $Fe_3O_4$ nanocubes with {400} facets were transformed into metallic Fe nanocubes with their surfaces terminated in {100} facets (Fig. 2b, c and Supplementary Fig. 6b, c). When metallic Fe nanocubes were exposed to the syngas, the thermodynamic driving force induced

the phase transition from metallic Fe to iron carbides. The surface Fe atoms underwent carburization and oxidation by reacting with carbon and oxygen species derived from the dissociated CO. After the exposure to syngas for 2 h, the corners were preferentially carburized and oxidized, resulting in the random distribution of $Fe_3O_4$ and χ-$Fe_5C_2$ domains (Fig. 2d). Notably, the diffractions of $Fe_3O_4$ and metallic Fe were clearly observed, whereas the crystalline χ-$Fe_5C_2$ was not detected by XRD (Supplementary Fig. 6d). This result implied that χ-$Fe_5C_2$ existed in short-range order with an amorphous structure. When Fe nanocubes were exposed to syngas for 5 h, carbon atoms cleaved from CO completely carburized the metallic Fe at the corner of the nanocube, while the dissociated oxygen atoms permeated into the interior, resulting in the bulk transition into $Fe_3O_4$ (Fig. 2e). At this stage, the diffractions of χ-$Fe_5C_2$ were observed in the XRD patterns, along with metallic Fe and $Fe_3O_4$ (Supplementary Fig. 6e). When Fe nanocubes were exposed to syngas for 10 h, the dissociated carbon atoms carburized the surface layers from the corners to the whole faces (Fig. 2f and Supplementary Fig. 6f). A core-shell structure with the $Fe_3O_4$ core and χ-$Fe_5C_2$ surface formed as the result of that the nanocubes lost the thermodynamic driving force for further carburization or oxidation[33]. The core-shell structure did not show obvious change when the syngas treatment was prolonged to 20 h (Fig. 1b, c). When the $Fe_3O_4$@χ-$Fe_5C_2$ core-shell structure formed at a steady state, the excess dissociated oxygen atoms were released into the gas phase in the form of $CO_2$ and $H_2O$ to prevent the oxidation of the shell. The excess dissociated carbon atoms reacted with surface-dissociated hydrogen atoms to yield hydrocarbons rather than to permeate and carburize the $Fe_3O_4$ interlayer. The stable core-shell structure was the result of the dynamic balance of the hydrocarbon production, surface oxidation, and carburization in the syngas environment.

We conclude the driving force to limit a single type of phase and facet at the surface layer as follows. The driving force to regulate the phase is the carbon chemical potential of reaction conditions. It was reported that the activation energy for carbon diffusion in Fe (43.9−69.0 kJ $mol^{-1}$) was lower than that for the FTS reaction (89.1 ± 3.8 kJ $mol^{-1}$)[14]. Consequently, surface carbon atoms cleaved from CO exhibit a pronounced affinity to Fe atoms, thereby instigating a phase transition from iron to $FeC_x$ during the initial stage of the FTS reaction. Upon the formation of active $FeC_x$ on the surface, the FTS reaction is facilitated, resulting in the release of oxygen species, predominantly in the form of $H_2O$, into the gas phase. This influx of $H_2O$ induces Fe oxide formation and impedes further carburization. As the concentration of oxygen species diminishes in the gas phase, a greater quantity of carbide accumulates on the surface, pushing the reaction condition back to the equilibrium position and vice versa. As the inner core of the catalyst oxidizes to $Fe_3O_4$, it becomes more resistant to carbon permeation and carburization compared to metallic Fe. According to carbon chemical potential theory, carbon-rich χ-$Fe_5C_2$ is the thermodynamically stable phase under our reaction condition (20 bar, CO:$H_2$ = 1:2, 270 °C)[21]. As such, other Fe carbides present in the surface layers will evolve into χ-$Fe_5C_2$ as the reaction progresses. Moreover, the uniform facet of the substrate and the small lattice mismatch (<5%) within the core-shell structure ensure the preservation of the epitaxial orientation relationship.

### Synthesis and characterization of $Fe_3O_4$@χ-$Fe_5C_2$ octahedra

For comparison, we applied the surface reconstruction procedure to prepare $Fe_3O_4$@χ-$Fe_5C_2$ octahedra. We carbonized the synthesized $Fe_3O_4$ octahedra following a similar approach to that of nanocubes (Supplementary Fig. 7). The average size of $Fe_3O_4$@χ-$Fe_5C_2$ octahedra was 45.4 ± 3.5 nm with a purity of 93.3% (Fig. 3a and Supplementary Fig. 8). The surface area of $Fe_3O_4$@χ-$Fe_5C_2$ octahedra was 22.0 $m^2 g^{-1}$ based on the BET method (Supplementary Fig. 4b). The lattice disparity between the core and the shell was clearly revealed by the HAADF-STEM images (Fig. 3b and Supplementary Fig. 9). The uniform

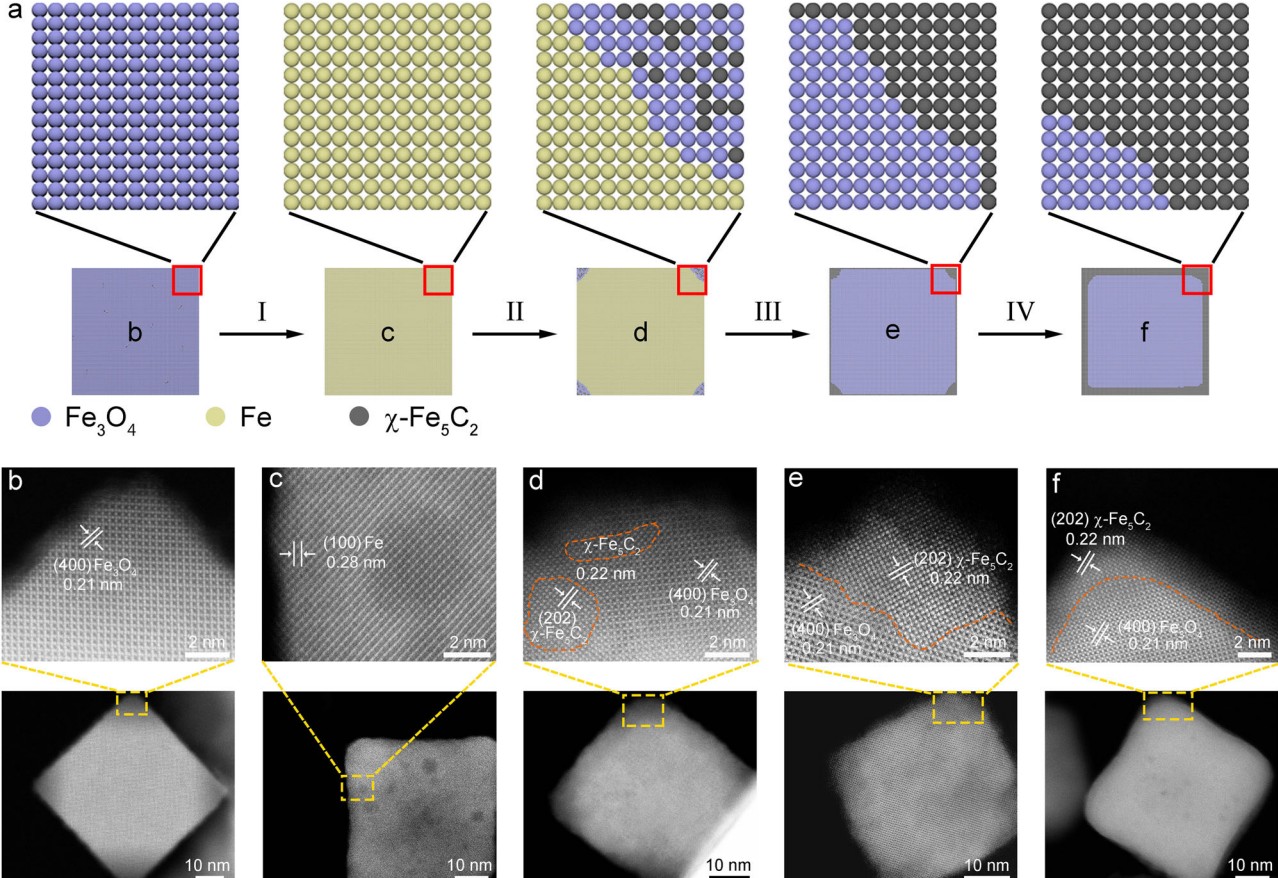

**Fig. 2 | Structural characterizations of the evolution from the Fe₃O₄ nanocube to the Fe₃O₄@χ-Fe₅C₂ nanocube. a** Schematic of the major steps involved in the continuous evolution from the Fe₃O₄ nanocube to the Fe₃O₄@χ-Fe₅C₂ nanocube. I refers to the reduction process; II refers to the corner carburization and oxidation; III refers to the surface carburization, O migration, and bulk oxidation; IV refers to the surface balanced carburization and oxidation. **b** HAADF-STEM images of an individual Fe₃O₄ nanocube. **c** HAADF-STEM images of an individual Fe nanocube. **d**–**f** HAADF-STEM images of an individual nanocrystal after the treatment of Fe nanocubes with syngas for 2, 5, and 10 h, respectively.

shell had an average thickness of 1.7 nm, corresponding to eight atomic layers (Fig. 3c). We assigned the lattice parameters with the help of the experimental and simulated FFT patterns. Specifically, the inner core took a lattice parameter of 0.48 nm which was indexed as the (111) facet of *fcc* Fe₃O₄ along the [001] zone axis (Fig. 3d, f, g). For the outer shell, the lattice parameter of 0.21 nm was assigned to the (112) facet of *bcm* χ-Fe₅C₂ along the [10-1] zone axis (Fig. 3e, h, i). Besides lattice matching, the epitaxial orientation can also be preserved via domain matching, where the spacing of *m* lattice planes in the epilayer is approximately equal to *n* in the substrate[32,34,35]. We observed that the spacing of three Fe₃O₄(111) planes in the core was approximately equal to that of seven χ-Fe₅C₂(112) planes in the shell (Fig. 3c). Such periodicity leads to a commensurate epitaxial relationship with a low mismatch value of 2.06% according to Eq. (2).

$$f = 2 \times |7 \times d_{shell} - 3 \times d_{core}|/(7 \times d_{shell} + 3 \times d_{core}) \qquad (2)$$

The domain match allows the conformal growth for the (112) facet of χ-Fe₅C₂ over the (111) facet of Fe₃O₄. The distribution of Fe₃O₄ at the core and χ-Fe₅C₂ at the shell was supported by the EELS image, STEM-EDX elemental mapping images, and line scanning profile (Fig. 3j–l). Mössbauer result indicated that Fe₃O₄@χ-Fe₅C₂ octahedra contained 65.1 wt% of Fe₃O₄ and 29.5 wt% of χ-Fe₅C₂ (Fig. 3m). The content of χ-Fe₅C₂ was consistent with the XRD quantitative analysis result (27.6%) (Supplementary Tables 1 and 2).

## Facet effect on activity and selectivity

We explored the facet effect of iron carbides on FTS properties. As the shells of Fe₃O₄@χ-Fe₅C₂ nanocrystals are more than six atomic layers thick, the electronic coupling between the core and outermost layer in the shell is essentially lost, and thus the ability to access the strain-dependent catalytic activity will be gone[36]. Besides, it was worth noting that no promoters or additives were added since the purpose of this work was to investigate the intrinsic catalytic performance of different exposed facets of χ-Fe₅C₂. Fe₃O₄@χ-Fe₅C₂ nanocubes and octahedra were loaded on the SiC support, denoted as Fe₃O₄@χ-Fe₅C₂ nanocubes/SiC and octahedra/SiC, respectively. The TEM images of these nanocrystals and the corresponding particle models from different orientations were shown in Supplementary Figs. 10 and 11. The catalytic properties of Fe₃O₄@χ-Fe₅C₂ nanocubes/SiC and octahedra/SiC were evaluated in a fixed-bed reactor under 20 bar of syngas (64 vol% H₂, 32 vol% CO, and 4 vol% Ar) with a space velocity of 2400 mL h⁻¹ g_cat⁻¹ at 270 °C, denoted as the standard condition. The CO conversion of Fe₃O₄@χ-Fe₅C₂ nanocubes/SiC was 45.4%, which was higher than that (21.2%) of Fe₃O₄@χ-Fe₅C₂ octahedra/SiC (Fig. 4a and Supplementary Table 3).

To compare the catalytic activity more accurately, we calculated the TOF numbers based on the moles of CO converted per mole of surface Fe atoms per hour. The moles of Fe atoms on the surface of Fe₃O₄@χ-Fe₅C₂ nanocrystals were determined by CO pulse chemisorption measurement. The moles of Fe atoms on the surface of Fe₃O₄@χ-Fe₅C₂ nanocubes/SiC was 24.1 μmol g⁻¹, higher than that

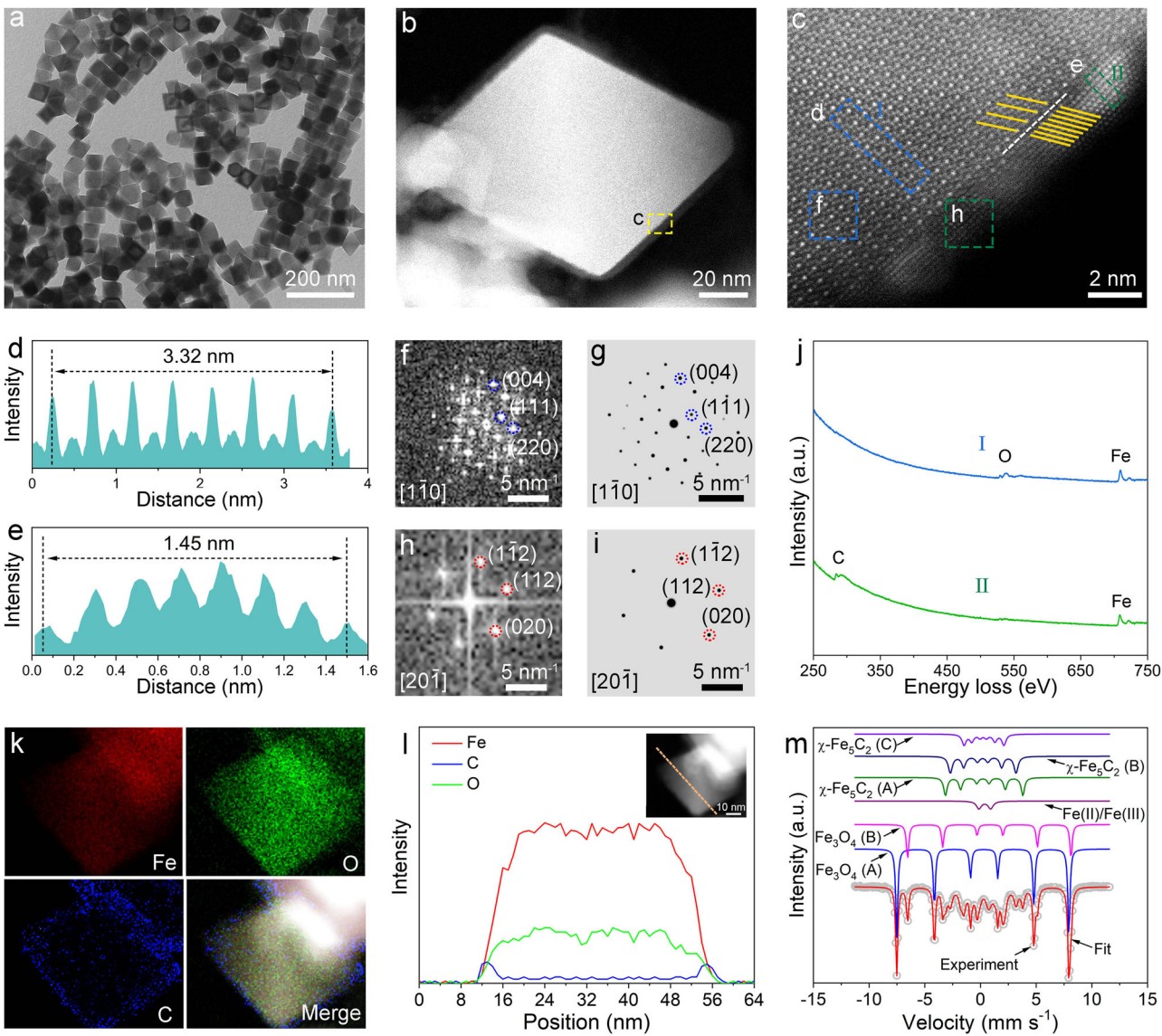

**Fig. 3 | Structural characterizations of Fe₃O₄@χ-Fe₅C₂ octahedra. a** TEM image of Fe₃O₄@χ-Fe₅C₂ octahedra. **b** HAADF-STEM image of an individual Fe₃O₄@χ-Fe₅C₂ octahedron. **c** Magnified HAADF-STEM image of the region marked by the corresponding box in panel (**b**). **d** Intensity profile recorded from the area indicated by the rectangular box in panel (**c**). **e** Intensity profile recorded from the area indicated by the rectangular box in panel (**c**). **f** FFT pattern from box **f** in panel (**c**). **g** Simulated FFT pattern from of Fe₃O₄ along the [1−10] direction. **h** FFT pattern from box **h** in panel (**c**). **g** Simulated FFT pattern from of χ-Fe₅C₂ along the [20−1] direction. **j** EELS spectra of a Fe₃O₄@χ-Fe₅C₂ octahedron in panel (**c**). **k** elemental mapping images of a Fe₃O₄@χ-Fe₅C₂ octahedron. **l** Line-scan profile recorded along the line of the inset HAADF image. **m** Mössbauer spectra of Fe₃O₄@χ-Fe₅C₂ octahedra.

(19.5 μmol g⁻¹) of Fe₃O₄@χ-Fe₅C₂ octahedra/SiC (Supplementary Fig. 12a, b). The TOF number of Fe₃O₄@χ-Fe₅C₂ nanocubes/SiC was 645.9 h⁻¹, being 1.7 times as high as that (372.7 h⁻¹) of Fe₃O₄@χ-Fe₅C₂ octahedra/SiC. The carbon balance value of Fe₃O₄@χ-Fe₅C₂ nanocubes/SiC was 98.7%, similar to that (96.5%) of Fe₃O₄@χ-Fe₅C₂ octahedra/SiC (Supplementary Table 3). The high carbon balance value indicated that the different CO conversions were not caused by continuous carburization. For reference, we prepared pure χ-Fe₅C₂ nanoparticles which mainly exposed the thermodynamically most stable {510} facet through the wet-chemical route[30] (Supplementary Fig. 13). The CO conversion of χ-Fe₅C₂ nanoparticles/SiC was 23.6% (Fig. 4a and Supplementary Table 3). As such, the {202} χ-Fe₅C₂ facet exposed on Fe₃O₄@χ-Fe₅C₂ nanocubes/SiC was more active than the {112} χ-Fe₅C₂ facet exposed on Fe₃O₄@χ-Fe₅C₂ octahedra/SiC and thermodynamically stable surfaces of iron carbides.

With respect to the selectivity, the C₅₊ selectivity of Fe₃O₄@χ-Fe₅C₂ nanocubes/SiC was 44.8 C%, higher than that (27.6 C%) of

Fe₃O₄@χ-Fe₅C₂ octahedra/SiC (Fig. 4a and Supplementary Table 3). The C₂-C₄ selectivity of Fe₃O₄@χ-Fe₅C₂ nanocubes/SiC was 41.0 C%, lower than that (53.1 C%) of Fe₃O₄@χ-Fe₅C₂ octahedra/SiC (Fig. 4a and Supplementary Table 3). Additionally, the ratio of olefins to paraffins (o/p ratio) among C₂-C₄ for Fe₃O₄@χ-Fe₅C₂ nanocubes/SiC was 1.1, higher than that (0.6) for Fe₃O₄@χ-Fe₅C₂ octahedra/SiC (Fig. 4a and Supplementary Table 3). The selectivity for C₂-C₄⁼ olefins over Fe₃O₄@χ-Fe₅C₂ nanocubes/SiC was 21.6 C%, approaching that (20.6 C%) over Fe₃O₄@χ-Fe₅C₂ octahedra/SiC (Supplementary Table 3). Additionally, the selectivity for C₅–C₁₂⁼ olefins over Fe₃O₄@χ-Fe₅C₂ nanocubes/SiC was 17.9 C%, higher than that (11.3 C%) over Fe₃O₄@χ-Fe₅C₂ octahedra/SiC (Supplementary Table 3). As for χ-Fe₅C₂ nanoparticles/SiC, the selectivities for CH₄, C₂-C₄, C₅-C₁₂, and C₁₃₊ were 15.5 C%, 48.2 C%, 34.9 C%, and 1.4 C%, respectively (Fig. 4a and Supplementary Table 3). The C₂-C₄⁼ and C₅-C₁₂⁼ selectivities were 29.8 C% and 16.9 C%, respectively (Supplementary Table 3). Actually, the distribution of hydrocarbon products for Fe₃O₄@χ-Fe₅C₂ nanocubes/SiC,

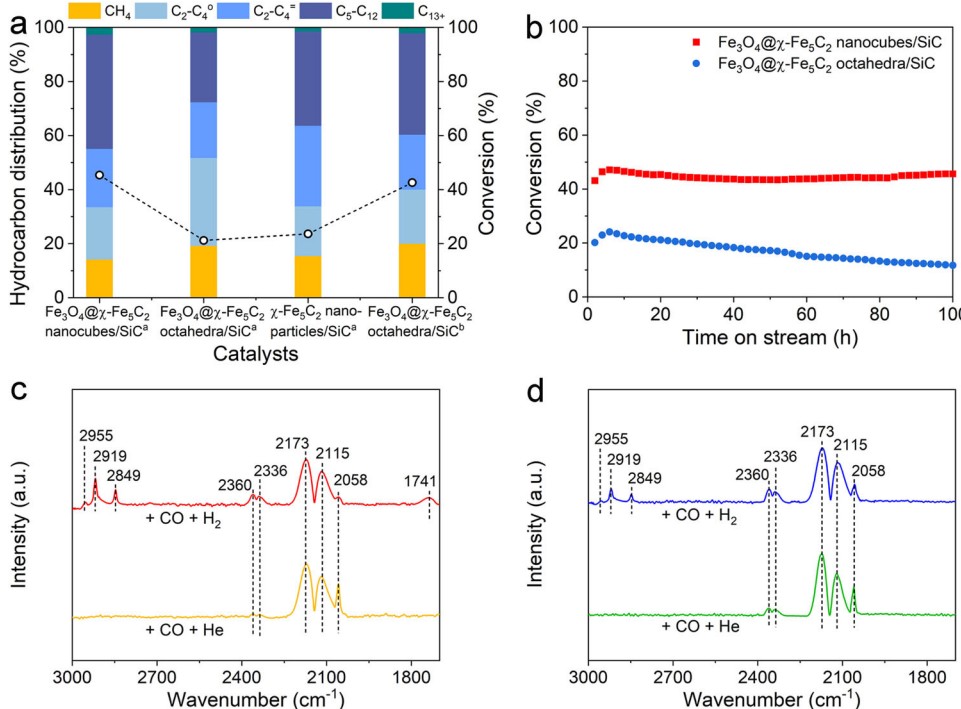

**Fig. 4 | Catalytic properties and structural characterizations of Fe₃O₄@χ-Fe₅C₂ nanocubes/SiC and octahedra/SiC. a** CO conversion and selectivity of Fe₃O₄@χ-Fe₅C₂ nanocubes/SiC, Fe₃O₄@χ-Fe₅C₂ octahedra/SiC, and χ-Fe₅C₂/SiC. ᵃ refers to that the reaction was conducted under 20 bar of syngas (CO:H₂ = 1:2, 2400 mL h⁻¹ g_cat⁻¹) at 270 °C. ᵇ refers to that the reaction was conducted under 20 bar of syngas (CO:H₂ = 1:2, 800 mL h⁻¹ g_cat⁻¹) at 270 °C. **b** Stability tests of Fe₃O₄@χ-Fe₅C₂ nanocubes/SiC and octahedra/SiC. The reaction was conducted under 20 bar of syngas (CO:H₂ = 1:2, 2400 mL h⁻¹ g_cat⁻¹) at 270 °C. **c** In-situ DRIFTS spectra of Fe₃O₄@χ-Fe₅C₂ nanocubes and **d** Fe₃O₄@χ-Fe₅C₂ octahedra after being exposed to CO for 30 min and purged with He or H₂ for 30 min at 270 °C.

Fe₃O₄@χ-Fe₅C₂ octahedra/SiC, and χ-Fe₅C₂ nanoparticles/SiC followed a typical Anderson-Schulz-Flory (ASF) statistics. The probabilities of chain growth ($\alpha$) for Fe₃O₄@χ-Fe₅C₂ nanocubes/SiC, Fe₃O₄@χ-Fe₅C₂ octahedra/SiC, and χ-Fe₅C₂ nanoparticles/SiC were calculated as 0.66, 0.62, and 0.63, respectively (Supplementary Figs. 14, 15). To compare the catalytic selectivity more appropriately, we adjusted the space velocity of Fe₃O₄@χ-Fe₅C₂ octahedra/SiC to keep the reaction over cubic and octahedral nanocrystals at similar conversion levels. When the space velocity was lowered to 800 mL h⁻¹ g_cat⁻¹, CO conversion of Fe₃O₄@χ-Fe₅C₂ octahedra/SiC reached 42.6%, approaching that (45.4%) of the cubic counterpart under the standard condition (Fig. 4a and Supplementary Table 3). The C₅₊ selectivity of Fe₃O₄@χ-Fe₅C₂ octahedra/SiC was 39.6 C%, while the C₂-C₄ hydrocarbons occupied 40.3 C% of all hydrocarbons (Fig. 4a and Supplementary Table 3). The selectivities for C₂-C₄= and C₅-C₁₂= were 18.3 C% and 16.0 C%, respectively (Supplementary Table 3) The $\alpha$ value was calculated as 0.64 at this conversion level (Supplementary Fig. 16). Therefore, the effect of crystal face on selectivity is not as great as that on activity.

We evaluated the long-term stability of Fe₃O₄@χ-Fe₅C₂ nanocrystals/SiC through a 100-h test. The CO conversion of Fe₃O₄@χ-Fe₅C₂ nanocubes/SiC fluctuated within 1% during the whole test, indicating the high stability of this catalyst (Fig. 4b). The cubic morphology was preserved after 100 h on stream (Supplementary Fig. 17a, b). We measured the thickness of the shell layer for Fe₃O₄@χ-Fe₅C₂ nanocubes after the reaction. The thickness of the shell increased from 2.0 nm to 2.6 nm after the reaction (Fig. 1c and Supplementary Fig. 17c). The lattice parameter of the Fe₃O₄ inner core was measured as 0.21 nm, which was indexed as the (400) facet of Fe₃O₄ (Supplementary Fig. 17d). The lattice parameter of the χ-Fe₅C shell was 0.22 nm, which was assigned to the (202) facet of χ-Fe₅C₂ (Supplementary Fig. 17e). The exposed χ-Fe₅C₂ facets of Fe₃O₄@χ-Fe₅C₂ nanocubes were preserved after 100 h on stream. The EELS image implied that the core region mainly comprised Fe and O elements while the shell region contained Fe and C elements (Supplementary Fig. 17f). We also conducted Mössbauer spectroscopy to characterize the compositions of the iron phase in the used Fe₃O₄@χ-Fe₅C₂ nanocubes after 100 h on stream (Supplementary Fig. 18a). The content of χ-Fe₅C₂ increased from 33.2% to 39.8% after the reaction (Supplementary Table 4).

In contrast, the CO conversion of Fe₃O₄@χ-Fe₅C₂ octahedra/SiC declined continuously. The conversion was only 11.7% after 100 h, about half of that (20.1%) at the beginning (Fig. 4b). To investigate the reason for the deactivation, we characterized Fe₃O₄@χ-Fe₅C₂ octahedra/SiC after 100 h on stream. As shown in Supplementary Fig. 19a, b, the solid octahedra collapsed after the reaction. The formation of Fe₃O₄@χ-Fe₅C₂ octahedra with multiple voids was ascribed to the Kirkendall effect[37–39]. Specifically, as the blockage of active sites by long-chain hydrocarbons and non-graphitic carbon, Fe₃O₄@χ-Fe₅C₂ octahedra were gradually deactivated. Meanwhile, the carbon chemical potential of reaction conditions changed. The dynamic balance of the hydrocarbon production, surface oxidation, and carburization in the syngas environment was broken. This led to the diffusion of Fe atoms between the Fe₃O₄ core and χ-Fe₅C₂ shell. The void formation in Fe₃O₄@χ-Fe₅C₂ octahedra due to differential diffusion rates of Fe atoms. The thickness of the shell increased from 1.7 nm to 3.5 nm after the reaction (Fig. 3c and Supplementary Fig. 19c). The lattice parameters of the inner core and the outer shell were measured as 0.48 nm and 0.21 nm, which were assigned to the (111) facet of Fe₃O₄ and (112) facet of χ-Fe₅C₂, respectively (Supplementary Fig. 19d, e). The exposed χ-Fe₅C facet of Fe₃O₄@χ-Fe₅C₂ octahedra was preserved after the reaction. The distribution of Fe₃O₄ at the core and χ-Fe₅C₂ at the shell was supported by the EELS image (Supplementary Fig. 19f). The content of χ-Fe₅C₂ in Fe₃O₄@χ-Fe₅C₂ octahedra increased from 29.5% to 40.4% after 100 h on stream (Supplementary Fig. 18b and Supplementary Table 4). The thickness of the shell layer for both Fe₃O₄@χ-Fe₅C₂ nanocubes and octahedra increased after the reaction.

Therefore, the stabilities of $Fe_3O_4$@$\chi$-$Fe_5C_2$ nanocrystals/SiC were also affected by the exposed facets.

To investigate the textural properties, we carried out $N_2$ physisorption characterizations. The pore-diameter distributions were analyzed using the Barrett-Joyner-Halenda (BJH) method. As shown in (Supplementary Fig. 4c, d), the surface layer of both $Fe_3O_4$@$\chi$-$Fe_5C_2$ nanocubes and octahedra were not porous. The absence of pores at the surface layer was also confirmed by HAADF-STEM images (Figs. 1b and 3b). We also measured the textural properties of spent $Fe_3O_4$@$\chi$-$Fe_5C_2$ nanocubes and octahedra after 100 h on stream (Supplementary Fig. 20a, b). The surface layer of spent $Fe_3O_4$@$\chi$-$Fe_5C_2$ nanocubes remained nonporous after 100 h on stream (Supplementary Fig. 20c). In contrast, spent $Fe_3O_4$@$\chi$-$Fe_5C_2$ octahedra contained mesopores as revealed by the pore-diameter distribution and HAADF-STEM image (Supplementary Fig. 20d). The average mesopore diameter was determined as 16.0 nm by the BJH method (Supplementary Fig. 20d).

To investigate other underlying mechanisms for catalyst deactivation, we employed Raman spectroscopy to analyze the surfaces of $Fe_3O_4$@$\chi$-$Fe_5C_2$ nanocubes and octahedra after 100 h. The presence of peaks at 1330 $cm^{-1}$ indicated the existence of disordered carbon (D band), while those at 1592 $cm^{-1}$ signified graphite (G band) (Supplementary Fig. 21a, b). Significantly higher intensities of these peaks were observed on the surface of spent $Fe_3O_4$@$\chi$-$Fe_5C_2$ octahedra compared with spent nanocubes, suggesting a greater accumulation of deposited carbon on the octahedral surface (Supplementary Fig. 21a, b). For a more precise comparison, we conducted thermogravimetric analysis (TGA) under the $N_2$ atmosphere on both $Fe_3O_4$@$\chi$-$Fe_5C_2$ nanocubes and octahedra after the reaction. The weight loss between 200 and 500 °C was attributed to the removal of long-chain hydrocarbons from the surface, while the weight loss beyond 500 °C was associated with the loss of non-graphitic carbon. In the case of spent $Fe_3O_4$@$\chi$-$Fe_5C_2$ nanocubes, a weight loss of 6.2 wt% was observed (Supplementary Fig. 21c). Conversely, during TGA testing, spent $Fe_3O_4$@$\chi$-$Fe_5C_2$ octahedra exhibited a total weight loss of 13.5 wt% due to long-chain hydrocarbons and deposited carbon (Supplementary Fig. 21d). The higher residual weight of long-chain hydrocarbons and deposited carbon on the octahedral structure suggests a more pronounced blockage of active sites compared to nanocubes. Hence, we hypothesize that carbon deposition also contributes to the deactivation of $Fe_3O_4$@$\chi$-$Fe_5C_2$ octahedra.

**Mechanistic insights into the facet effect**

To rationalize the facet-dependent FTS activity, we explored CO dissociation pathways by conducting in-situ diffuse reflection infrared spectroscopy (DRIFTS) measurements at 270 °C. When $Fe_3O_4$@$\chi$-$Fe_5C_2$ nanocubes were exposed to 1 bar of CO and purged with He, three sets of peaks were observed (Fig. 4c). The peaks at 2173 and 2115 $cm^{-1}$ were assigned to the gaseous CO, while the peak at 2058 $cm^{-1}$ arose from the stretching vibration of linearly adsorbed CO[19,40–42]. The peaks at 2360 and 2336 $cm^{-1}$ corresponded to the gaseous $CO_2$, indicating the direct dissociation of CO on $Fe_3O_4$@$\chi$-$Fe_5C_2$ nanocubes. As for the exposure to CO and purging with $H_2$, other sets of peaks emerged besides the peaks for CO and gaseous $CO_2$ (Fig. 4c). Specifically, the peak at 2955 $cm^{-1}$ derived from the asymmetrical stretching vibration of C-H in $CH_3$* (refs. 43,44). The peaks at 2919 and 2849 $cm^{-1}$ were indexed as the asymmetrical and symmetrical stretching vibrations of C-H in CHO* and $CH_2$*, respectively[45,46]. Notably, the peak at 1741 $cm^{-1}$ was ascribed to the stretching vibration of C = O in CHO* species, which indicated the existence of a hydrogen-assisted dissociation route[47,48]. For $Fe_3O_4$@$\chi$-$Fe_5C_2$ octahedra, when the sample was exposed to CO and purged with He, the peaks for gaseous $CO_2$ appeared (Fig. 4d). As for the treatment with CO and purging with $H_2$, the in-situ DRIFTS profile showed the peaks for $CH_3$*, $CH_2$*, $CO_2$, and CO, in the absence of

CHO* or COH* (Fig. 4d). We also conducted in-situ DRIFTS experiments under 20 bar of syngas, simulating realistic reaction environments (Supplementary Fig. 22, Supplementary Table 5). The appearance of gaseous $CO_2$ peaks at 2360 and 2336 $cm^{-1}$ provided evidence for direct dissociation occurring on both $Fe_3O_4$@$\chi$-$Fe_5C_2$ nanocubes and octahedra (Supplementary Fig. 22, Supplementary Table 5). Moreover, the presence of CHO* species, as indicated by a peak at 1741 $cm^{-1}$, was observed solely on the $Fe_3O_4$@$\chi$-$Fe_5C_2$ nanocubes catalyst and absent on the octahedral counterpart (Supplementary Fig. 22, Supplementary Table 5). Therefore, $Fe_3O_4$@$\chi$-$Fe_5C_2$ octahedra enabled the direct dissociation of CO, while both direct and hydrogen-assisted CO dissociation routes existed on $Fe_3O_4$@$\chi$-$Fe_5C_2$ nanocubes (Supplementary Fig. 23).

To investigate the activation energy of different CO dissociation routes, we plotted the area of linearly adsorbed CO peak at 2058 $cm^{-1}$ as a function of time at 190, 230, and 270 °C (Supplementary Figs. 24–27). Based on the slope of the decrease in peak area with purge time, the rate of CO dissociation ($k$) was obtained. Activation energies of CO dissociation were calculated with the Arrhenius equation[7]. For $Fe_3O_4$@$\chi$-$Fe_5C_2$ nanocubes, the activation energy for hydrogen-assisted dissociation of CO was 68.6 kJ $mol^{-1}$, significantly lower than that (103.7 kJ $mol^{-1}$) direct dissociation of CO (Supplementary Fig. 28a). With regard to $Fe_3O_4$@$\chi$-$Fe_5C_2$ octahedra, the activation energy (83.2 kJ $mol^{-1}$) of CO dissociation with $H_2$ approximated to that (80.4 kJ $mol^{-1}$) without $H_2$ (Supplementary Fig. 28b). Therefore, $Fe_3O_4$@$\chi$-$Fe_5C_2$ nanocrystals exhibited facet-dependent FTS activities derived from the alteration of reaction paths which changed the activation energy. Specially, $Fe_3O_4$@$\chi$-$Fe_5C_2$ nanocubes followed hydrogen-assisted CO dissociation which lowered the activation energy, relative to that of direct CO dissociation over $Fe_3O_4$@$\chi$-$Fe_5C_2$ octahedra.

We conducted DFT calculations to rationalize CO direct dissociation route and hydrogen-assisted dissociation path on $\chi$-$Fe_5C_2$(202) and $\chi$-$Fe_5C_2$(112) surface. For $\chi$-$Fe_5C_2$(202) surface, the hydrogen-assisted CO dissociation route is the dominating route, since its energy barrier of the rate-limiting step (CO* + H* → HCO*) 1.23 eV is much lower than the direct CO dissociation route with an energy barrier as high as 2.85 eV (Supplementary Figs. 29–32). For $\chi$-$Fe_5C_2$(112) surface, the direct CO dissociation route exhibited a lower energy barrier (1.37 eV) than the hydrogen-assisted CO dissociation route (1.54 eV), implying the direct dissociation as the main route of CO dissociation (Supplementary Figs. 33–36). Besides, the energy barrier of the hydrogen-assisted CO dissociation route on the $\chi$-$Fe_5C_2$(202) surface is lower than that of the CO direct dissociation on the $\chi$-$Fe_5C_2$(112) surface (Supplementary Figs. 32 and 35). The DFT conclusion was consistent with in-situ DRIFTS spectra result.

We explored the adsorption of CO and $H_2$ by conducting pulse chemisorption measurements to rationalize why $Fe_3O_4$@$\chi$-$Fe_5C_2$ nanocubes/SiC exhibited better carbon-chain growth ability. The amount of adsorbed gas was calculated on the difference between the total amount of gas injected and the amount measured at the outlet from the sample. The amount of adsorbed CO over $Fe_3O_4$@$\chi$-$Fe_5C_2$ nanocubes/SiC was 24.1 μmol $g^{-1}$, higher than that (19.5 μmol $g^{-1}$) of $Fe_3O_4$@$\chi$-$Fe_5C_2$ octahedra (Supplementary Fig. 12). While the amount of adsorbed $H_2$ over $Fe_3O_4$@$\chi$-$Fe_5C_2$ nanocubes/SiC was 7.3 μmol $g^{-1}$, lower than that (11.0 μmol $g^{-1}$) of the octahedral counterpart (Supplementary Fig. 37). The results indicated that $Fe_3O_4$@$\chi$-$Fe_5C_2$ nanocubes/SiC improved the CO adsorption and suppressed the $H_2$ adsorption compared to the octahedral counterpart. Thus, the surface CO/$H_2$ ratio of $Fe_3O_4$@$\chi$-$Fe_5C_2$ nanocubes/SiC was 3.3, higher than that (1.8) of the octahedral counterpart (Supplementary Fig. 38). With the increase of the surface CO/$H_2$ ratio, the $CH_4$ production over $Fe_3O_4$@$\chi$-$Fe_5C_2$ nanocubes/SiC was suppressed, while hydrocarbon products shifted to long-chain hydrocarbons. We also conducted DFT

calculations to investigate the facet effect on $CH_4$ production. The $CH_2^* + H^*$ energy barrier of $\chi$-$Fe_5C_2$(202) is 1.03 eV, higher than that (0.70 eV) of $\chi$-$Fe_5C_2$(112) (Supplementary Figs. 39–41). The energy barrier of the $CH_2^* + CH_2^*$ step over $\chi$-$Fe_5C_2$(202) facet is 0.70 eV, lower than that (1.03 eV) of $CH_2^* + H^*$. As for $\chi$-$Fe_5C_2$(112) facet, the energy barrier of the $CH_2^* + CH_2^*$ step is 0.60 eV, approaching to that (0.7 eV) of $CH_2^* + H^*$. Thus, compared with $Fe_3O_4$@$\chi$-$Fe_5C_2$ nanocubes, the octahedral counterpart benefits the $CH_4$ production.

We demonstrated the dependence of catalytic performance on the exposed facet of iron carbides in FTS. Uniformly exposed {202} and {112} facets of $\chi$-$Fe_5C_2$ were successfully fabricated as the model system. With the help of well-defined catalysts, we identified the intrinsically active, selective, and stable facets of $\chi$-$Fe_5C_2$. Our findings deepen the understanding of Fe-based FTS catalysts from the phase and size to the facet. In addition, this work also provides a facile method to precisely control the exposed surfaces of iron carbides for both future fundamental studies and practical applications.

## Methods

### Chemicals and materials

Oleylamine (OAm, >70%), oleic acid (OA, 90%), benzyl ether (BE, 99%), iron(III) acetylacetonate (Fe(acac)$_3$, >99.9%), and 4-biphenylcarboxylic acid (99%) were obtained from Sigma-Aldrich. SiC ($\beta$-phase, 99.8%) was obtained from Adamas-beta®. All other chemicals were analytical grade and purchased from Sinopharm Chemical Reagent Co., Ltd. Deionized water with a resistivity of 18.2 MΩ cm was used for the preparation of all aqueous solutions (Milli-Q®).

### Synthesis of $Fe_3O_4$ nanocrystals

In a typical synthesis of $Fe_3O_4$ nanocubes, Fe(acac)$_3$ (1.4 g) and 4-biphenylcarboxylic acid (1.0 g) were dissolved in a mixture of BE (20.0 mL) and OA (2.5 mL). The solution was degassed at 120 °C for 30 min. Subsequently, the solution was heated to 290 °C at 10 °C min⁻¹ and kept for 30 min with a stirring rate of 300 rpm. As for the synthesis of $Fe_3O_4$ octahedra, Fe(acac)$_3$ (1.0 g) was dissolved in a mixture of BE (20.0 mL), OAm (2.3 mL), and OA (1.6 mL), followed by degassing at 120 °C for 30 min. The solution was heated to 220 °C at 20 °C min⁻¹ with a stirring rate of 300 rpm and kept for 1 h. Afterward, the solution was heated to 300 °C at 20 °C min⁻¹ and kept for 2 h. After the solution had been cooled down to room temperature, the products were precipitated by ethanol, washed three times with hexane, and re-dispersed in hexane.

### Synthesis of $Fe_3O_4$@$\chi$-$Fe_5C_2$ nanocrystals

For the synthesis of $Fe_3O_4$@$\chi$-$Fe_5C_2$ nanocrystals, as-prepared $Fe_3O_4$ nanocrystals (50 mg) were loaded into a fixed-bed reactor with an inner diameter of 9 mm. $Fe_3O_4$ nanocrystals were reduced in $H_2$ under 1 bar with a gas-flow rate of 100 mL min⁻¹ at 270 °C for 10 h, while the heating ramp was 1 °C min⁻¹. Afterwards, the obtained samples underwent surface reconstruction under 20 bar of syngas (32 vol% $H_2$, 64 vol% CO, and 4 vol% Ar) with a space velocity of 2400 mL h⁻¹ g$_{cat}$⁻¹ at 270 °C for 20 h. Considering that the iron carbides were highly sensitive to the atmosphere, we used inert gas to protect and store the catalyst after reaction. Specifically, we switched the feed gas to $N_2$ when the reaction was stopped. The reaction tubes were sealed via valves at both ends after cooling the reactor to room temperature. Afterwards, the reaction tubes were transferred to a $N_2$-filled glove box. The samples were stored in the glove box before characterizations.

### Synthesis of pure $\chi$-$Fe_5C_2$ nanocrystals

Octadecylamine of 14.5 g and cetyl trimethyl ammonium bromide (CTAB) of 0.113 g were mixed in a four-neck flask under stirring and degassed under the $N_2$ flow, followed by being heated to 120 °C.

Afterwards, Fe(CO)$_5$ (3.6 mmol) was injected into the mixture under the $N_2$ blanket. The mixture was heated to 180 °C at 10 °C min⁻¹ and kept at this temperature for 10 min. Subsequently, the mixture was further heated to 350 °C at 10 °C min⁻¹ and kept at this temperature for 10 min. After being cooled to room temperature, the products were washed with a mixture of ethanol and hexane.

### Mössbauer measurements

$^{57}$Fe Mössbauer spectra were carried out on a Topologic 500 A spectrometer driving with a proportional counter at room temperature. The radioactive source was $^{57}$Co (Rh) moving in a constant acceleration mode. Data analyses were performed assuming a Lorentzian lineshape for computer folding and fitting.

### Catalytic tests

The Fischer-Tropsch reaction was carried out in a fixed-bed reactor under 20 bar of syngas at 270 °C. Generally, $Fe_3O_4$@$\chi$-$Fe_5C_2$ nanocrystals (50 mg, 20–40 meshes) were diluted with SiC (450 mg, 20–40 meshes). The sample was loaded into a fixed-bed reactor with an inner diameter of 9 mm. Subsequently, a mixture including 96 vol% of $H_2$/CO mixed gas (64 vol% $H_2$, 32 vol% CO) and 4 vol% of Ar (as an internal standard) was introduced to the reactor as the feeding gas at a space velocity of 2400 mL h⁻¹ g$_{cat}$⁻¹.

The gaseous products were monitored by online gas chromatographs (Shimadzu GC-2014). An ice trap with 2.0 g of solvent (n-dodecane) was employed to capture the liquid hydrocarbons in the effluent. The liquid products with 0.1 g of an internal standard (decalin) were analyzed using an offline Shimadzu GC-2014.

CO conversion was calculated as follows:

$$\text{CO conversion} = \frac{\text{CO}_{\text{inlet}} - \text{CO}_{\text{outlet}}}{\text{CO}_{\text{inlet}}} \times 100\% \qquad (3)$$

where $CO_{inlet}$ and $CO_{outlet}$ are moles of CO at the inlet and outlet, respectively.

$CO_2$ selectivity was calculated according to:

$$\text{CO}_2 \text{ selectivity} = \frac{\text{CO}_{\text{outlet}}}{\text{CO}_{\text{inlet}} - \text{CO}_{\text{outlet}}} \times 100\% \qquad (4)$$

where $CO_{2\text{ outlet}}$ refers to moles of $CO_2$ at the outlet.

The selectivity of hydrocarbon $C_nH_m$ was obtained according to:

$$\text{C}_n\text{H}_m \text{ selectivity} = \frac{n\text{C}_n\text{H}_{m\text{ outlet}}}{\sum_i i\text{C}_i\text{H}_{m\text{ outlet}}} \times 100\% \qquad (5)$$

where $C_nH_{m\text{ outlet}}$ represents moles of individual hydrocarbon product at the outlet.

Carbon balance was calculated according to:

$$\text{Carbon balance} = \frac{\sum_i i\text{C}_i\text{H}_{m\text{ outlet}} + \text{CO}_2}{\text{CO}_{\text{inlet}} - \text{CO}_{\text{outlet}}} \times 100\% \qquad (6)$$

where $C_nH_m$ and $CO_2$ represent the moles of the produced hydrocarbons and $CO_2$, respectively. $CO_{inlet}$ and $CO_{outlet}$ are moles of CO at the inlet and outlet, respectively. The carbon balances were over 95.0%.

### In-situ DRIFTS measurements after different gas treatments

In-situ DRIFTS experiments were conducted in an elevated-pressure cell (DiffusIR Accessory PN 041-10XX) with a Fourier transform infrared spectrometer (TENSOR II Sample Compartment RT-DLaTGS) and a liquid-nitrogen-cooled MCT detector. The outlet gas was analyzed by a mass spectrometer (Hiden HPR20). Spectra were measured accumulating 64 scans at a resolution of 4 cm⁻¹. Prior to the

test, the samples were cleaned in He with a gas-flow rate of 100 mL min⁻¹ at 270 °C for 1 h. Then the samples were exposed to 1 bar of CO for 30 min and subsequently purged by He or $H_2$ for 30 min at 270 °C. As for the in-situ DRIFTS measurements at different temperatures, the background spectra of the samples were acquired under He after the temperature of the samples dropped to a specified temperature. Then, the samples were exposed to 1 bar of CO for 30 min and subsequently purged with He or $H_2$ at a specified temperature. As for the in-situ DRIFTS experiments under 20 bar of syngas, thorough cleaning of the samples in He at 270 °C for 1 h ensured the elimination of any contaminants. Background spectra were acquired under He flow, followed by exposure to a 20-bar syngas mixture (CO/$H_2$) at 270 °C for 30 min.

## DFT methods

Spin-polarized DFT calculations were conducted using the Vienna ab initio simulation package (VASP)[49]. The projector-augmented wave method and the Perdew-Burke-Ernzerhof functional were implemented in the code[50]. $Fe_5C_2$(202) and $Fe_5C_2$(112) slab were established with the same stoichiometric atoms ($Fe_{60}C_{24}$). A plane-wave basis set with a cutoff energy of 400 eV was used with the K-points of $3 \times 3 \times 1$ and vacuum thickness of 15 Å. During the structure optimization, the two layers of atoms at the bottom were fixed, while the others include the adsorbates were relaxed with the electronic convergence of 0.02 eV/Å. For each elementary step, the initial states and final states are firstly optimized. The transition states are searched using the climbing image nudged elastic band method (CI-NEB) and confirmed by the vibrational frequencies analysis[51].

## CO pulse chemisorption measurements

CO pulse chemisorption measurements were performed using a Micromeritics Autochem 2920 chemisorption analyzer with an active loop volume of 0.1 mL. In a typical measurement, 100 mg of $Fe_3O_4$@χ-$Fe_5C_2$ nanocrystals/SiC were packed into a reactor with a quartz tube. Prior to the test, the samples were cleaned in He with a gas-flow rate of 100 mL min⁻¹ at 270 °C for 5 h. After cooling down to 50 °C under He flow, CO/He pulses (10 vol% CO and 90 vol% He) were injected until adsorption reached saturation. The amount of adsorbed CO was calculated on the difference between the total amount of CO injected and the amount measured at the outlet from the sample. The metal dispersion was calculated by assuming the ratio of CO to surface metal atom as 1:1.

TOF number was calculated according to:

$$\begin{aligned}TOF = {} & CO\ conversion \times moles\ of\ CO\ in\ syngas \times gas \\ & - flow\ rate \div moles\ of\ surface\ Fe\ atoms\end{aligned} \tag{7}$$

## H₂ pulse chemisorption measurements

$H_2$ pulse chemisorption measurements were performed using a Micromeritics Autochem 2920 chemisorption analyzer with an active loop volume of 0.1 mL. In a typical measurement, 100 mg of $Fe_3O_4$@χ-$Fe_5C_2$ nanocrystals/SiC were packed into a reactor with a quartz tube. Prior to the test, the samples were cleaned in He with a gas-flow rate of 100 mL min⁻¹ at 270 °C for 5 h. After cooling down to 50 °C under He flow, $H_2$/Ar pulses (10 vol% $H_2$ and 90 vol% Ar) were injected until adsorption reached saturation. The amount of adsorbed $H_2$ was calculated on the difference between the total amount of $H_2$ injected and the amount measured at the outlet from the sample.

## Instrumentations

TEM images were taken using a Hitachi H-7700 transmission electron microscope at an acceleration voltage of 100 kV. HAADF and EDS analysis were collected on a JEOL ARM-200F field-emission transmission electron microscope operating at 200 kV accelerating voltage. XPS measurements were conducted on an ESCALAB 250 (Thermo-VG Scientific, USA) with an Al Kα X-ray source (1486.6 eV protons) in Constant Analyzer Energy (CAE) mode with a pass energy of 30 eV for all spectra. XRD characterization was performed using a Philips X'Pert Pro X-ray diffractometer with a monochromatized Cu Kα radiation source and a wavelength of 0.1542 nm. BET measurements were carried out on Micromeritics AutoChem II 2020. TGA spectra were conducted on Pyris Diamond TG-DTG. Raman spectra were detected by a Renishaw RM3000 Micro-Raman system with a 514.5 nm Ar laser.

## Data availability

The data generated in this study are provided in the Supplementary Information.

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

## Acknowledgements

This work was supported by National Key Research and Development Program of China (2023YFA1508003 to H.L., 2021YFA1500500 to J.Z., 2019YFA0405600 to J.Z.), CAS Project for Young Scientists in Basic Research (YSBR-051 to J.Z.), National Science Fund for Distinguished Young Scholars (21925204 to J.Z.), NSFC (22221003 to J.Z., 22250007 to J.Z., 22361162655 to J.Z., 22204158 to W.W., 22308346 to H.L.), Fundamental Research Funds for the Central Universities, Collaborative Innovation Program of Hefei Science Center, CAS (2022HSC-CIP004 to J.Z.), the Joint Fund of the Yulin University and the Dalian National Laboratory for Clean Energy (YLU-DNL Fund 2022012 to J.Z.), USTC Research Funds of the Double First-Class Initiative (YD9990002014 to H.L.), Joint Funds from the Hefei National Synchrotron Radiation Laboratory (KY2340000157 to W.W., KY9990000202 to H.L.), the DNL Cooperation Fund, CAS (DNL202003 to J.Z.), and International Partnership Program of Chinese Academy of Sciences (123GJHZ2022101GC to J.Z.). J.Z. acknowledges support from the Tencent Foundation through the XPLORER PRIZE. This work was partially carried out at the Instruments Center for Physical Science, University of Science and Technology of China. This work was also partially carried out at the USTC Center for Micro and Nanoscale Research and Fabrication.

## Author contributions

W.W. and J.L. equally contributed to this work. W.W., J.L., H.L., and J. Zeng designed the studies and wrote the paper. W.W., J.L., and L.L. synthesized catalysts. W.W., J.L., B.H., and H.L. performed catalytic tests. W.W. and W.M. conducted characterizations. J. Zhao conducted DFT calculations. S.H. and C.M. conducted HAADF-STEM characterizations. All authors discussed the results and commented on the paper.

## Competing interests

The authors declare no competing interests.
