## [Peer Review File · Nature Communications]

Facet sensitivity of iron carbides in Fischer-Tropsch synthesisREVIEWER COMMENTS

Reviewer #1 (Remarks to the Author):

This work prepares varied tailor-made FeOx@FeCx capsule-like catalysts for Fischer-Tropsch synthesis (FTS), where surface layer is limited to one type of Fe carbide phase or facet. As in general, many kinds of Fe carbide and facet co-exist in conventional FTS catalysis, it was rather difficult to make clear the different role and function of each Fe carbide phase and facet type, until now. This paper determines and realizes clear catalyst structure being oriented to solve this vital FTS problem. The findings such as different facets exhibiting varied activity but similar chain-growth probability are interesting and important. I recommend the publication of this paper after suitable revision.

(1) What is the driving force to regulate or limit one type of facet or phase at the surface layer? Please provide more insights on the conformal growth here.

(2) The catalyst structure before and after the reaction should be compared in detail. Does thickness of the layer change?

(3) Is the surface layer (FeCx) porous?

(4) TOFs of nanocubes and octahedra are not the same?

(5) More info on the products is expected (i.e. olefinic hydrocarbon selectivity).

(6) Some sentences are need to be improved. For example, "0.113g of CTAB" should be "CTAB of 0.113g".

Reviewer #2 (Remarks to the Author):

The authors successfully fabricated two core-shell catalysts with {202} and {112} facets of χ -Fe₅C₂ as the outer shell through the conformal reconstruction of Fe₃O₄ nanocubes and octahedra, as the inner cores. The sensitivity of the facet of iron carbides to performance were explored. The different types of CO dissociation led to different FTS activity. Although some interesting results have been got, there are still some problems need to be addressed.

1. First and most importantly, as we know, the carbon permeation, diffusion and carburization occur. Therefore, the Fe-based catalysts undergo a long activation period before the structure is stable. In this work, the authors correlated the activity to the different facets. But no solid evidence confirms that the stable structure has been achieved. The carbon balance data were not provided. So, we can confirm that the different CO conversions are assigned to different activity or continuous carburization or structure evolution. And the CO/H₂ ratio during activation and reaction is different. Even the TEM images for the used catalysts were provided, the Fe₃O₄@ χ -Fe₅C₂ octahedra collapsed, and the nanocrystals seem to be more stable. But is the thickness of the carbide shell the same? Is the exposed facet maintained after reaction? And also, the compositions of iron phase of the used catalysts are missed. This is very critical for the main conclusion of this work. Because it is very common that the "apparent activity" keeps a long time, but the structure of carbides changes a lot.

2. The authors claimed that the facets are not sensitive for carbon-chain growth. But the CH₄ production is obviously improved. Is there any explanation?

3. The pressure of DRIFTS is lower than the reaction evaluation. More important, the H₂ is induced after CO adsorption in DRIFTS experiments. This is very different from the real reaction that the CO and H₂ are co-fed. The hydrogen-assisted CO dissociation usually occurs in the co-existence of CO and H₂. And even in the results of this work, once CO adsorbed, the dissociation occurred on both octahedra and nanocube. So, how do the authors discriminate the two kinds of CO dissociation?

4. I think if the DFT calculations are given, the conclusion will be more supportive.

Reviewer #3 (Remarks to the Author):

In this work, the authors constructed Fe₃O₄@Fe₅C₂ core-shell with different facets which exhibited comparable activity and catalytic mechanism. However, the reconstruction of Fe₅C₂ during Fischer-Tropsch reaction has been systematically investigated in early reported works. (J. Phys. Chem. C 2017, 121, 9, 5154-5160; ACS Catal. 2017, 7, 9, 5661-5667) The synthesis of Fe₃O₄@Fe₅C₂ core-shell for Fischer-Tropsch synthesis has also been reported. (ACS Catal. 2016, 6, 6, 3610-3618) The catalytic mechanism has also been reported. (Applied Energy 2015, 160, 15, 982-989). Therefore, I think the authors should really differentiate their works in novelty from the literature reports before getting publishing on Nat. Commun..

Some specific points:

1. In page 8 line 221, the authors claimed that "the solid octahedra collapsed after the reaction." However, In Fig. S16, it is easy to find the collapsed nanocubes while the CO conversion was nearly constant. The difference of stability of two samples could not be only contributed to the collapsed morphology.
2. Schematic diagram for different Fischer-Tropsch mechanisms of Fe₃O₄@Fe₅C₂ with different facets should be provided to help the readers to understand the molecular catalytic mechanisms.

Point-by-point response to reviewers' comments

Manuscript ID: NCOMMS-23-62544

Title: Facet sensitivity of iron carbides in Fischer-Tropsch synthesis

Reviewer #1

“This work prepares varied tailor-made FeO_x@FeC_x capsule-like catalysts for Fischer-Tropsch synthesis (FTS), where surface layer is limited to one type of Fe carbide phase or facet. As in general, many kinds of Fe carbide and facet co-exist in conventional FTS catalysis, it was rather difficult to make clear the different role and function of each Fe carbide phase and facet type, until now. This paper determines and realizes clear catalyst structure being oriented to solve this vital FTS problem. The findings such as different facets exhibiting varied activity but similar chain-growth probability are interesting and important. I recommend the publication of this paper after suitable revision.”

We sincerely thank this reviewer's valuable comments on our work. We have clarified the comments raised by this reviewer as follows.

“(1) What is the driving force to regulate or limit one type of facet or phase at the surface layer? Please provide more insights on the conformal growth here.”

We sincerely thank this reviewer for his/her valuable comment. To achieve the specific facet and phase, our approach began with the synthesis of Fe₃O₄ templates with uniform size and facet. We finely controlled the shape by selectively stabilizing crystal facets through capping ligands and modulating growth kinetics. When 4-biphenylcarboxylic acid was used to bind {100} surfaces, Fe₃O₄ nanocubes were obtained. When oleylamine and stepped temperature programs were adopted, Fe₃O₄ octahedra were formed. After the *in-situ* reduction process, we obtained Fe nanocubes and octahedra.

The driving force to regulate one type of phase at the surface layer of Fe₃O₄@ χ -Fe₅C₂ nanocrystals is the carbon chemical potential of reaction conditions. It was reported that the activation energy for carbon diffusion in Fe (43.9-69.0 kJ mol⁻¹) was lower than that for the FTS reaction (89.1±3.8 kJ mol⁻¹) [*Chem. Soc. Rev.* **37**, 2758-81 (2008)]. Consequently, surface carbon atoms cleaved from CO exhibit a pronounced affinity to Fe atoms, thereby instigating a phase transition from iron to FeC_x during the initial stage of the FTS reaction. Upon the formation of active FeC_x on the surface, the FTS reaction is facilitated, resulting in the release of oxygen species, predominantly in the form of H₂O, into the gas phase. This influx of H₂O induces Fe oxide formation and impedes further carburization. As the concentration of oxygen species diminishes in the gas phase, a greater quantity of carbide accumulates on the surface, pushing the reaction condition back to the equilibrium position and vice versa. As the inner core of the catalyst oxidizes to Fe₃O₄, it becomes more resistant to carbon permeation and carburization compared to metallic Fe. When the reaction reaches equilibrium, the catalysts ultimately transform into a core-shell structure consisting of bulk Fe₃O₄ with surface FeC_x. According to carbon chemical potential theory, carbon-rich χ -Fe₅C₂ is the thermodynamically stable phase under our reaction condition (20 bar, CO:H₂ = 1:2, 270 °C) [*J. Am. Chem. Soc.* **132**, 14928-14941 (2010)]. As such, other Fe carbides present in the surface layers will evolve into χ -Fe₅C₂ as the reaction progresses. The uniform size and facets of Fe₃O₄ templates ensure a consistent rate of evolution. Consequently, only a single type of phase was observed at the surface layer over Fe₃O₄@ χ -Fe₅C₂ nanocrystals under equilibrium reaction conditions.

The small lattice mismatch results in only one type of facet at the surface layer of Fe₃O₄@ χ -Fe₅C₂ nanocubes. In the case of nanocrystals with a core-shell structure, a slight lattice mismatch ($f < \sim 5\%$) is required for the epitaxial surface layer to form over the inner core [Chem. Rev. **120**, 2123-2170 (2020)]. This epitaxial relationship allows for the maintenance of orientation between the growth layer and the substrate within the first few atomic layers. The spacing of Fe₃O₄(400) planes (0.21 nm) in the core was approximately equal to that of χ -Fe₅C₂(202) planes (0.22 nm) in the shell (Fig. R1a). The lattice mismatch of Fe₃O₄@ χ -Fe₅C₂ nanocubes was calculated as 4.65% (equation 1), ensuring the preservation of the epitaxial orientation relationship.

$$f = 2 \times |d_{\text{shell}} - d_{\text{core}}| / (d_{\text{shell}} + d_{\text{core}}) \quad (1)$$

In equation 1, d_{shell} and d_{core} refer to the lattice spacings of the shell and the core, respectively. Due to the close match of lattice constants, atomic smooth Fe₃O₄@ χ -Fe₅C₂ nanocubes can be readily accessed via conformal growth, where the (202) facet of χ -Fe₅C₂ forms over the (400) facet of Fe₃O₄.

Besides lattice matching, the epitaxial orientation can also be preserved via domain matching, where the spacing of m lattice planes in the epilayer is approximately equal to n in the substrate [Nano Lett. **10**, 3028-3036 (2010); Nanoscale **10**, 9862-9866 (2018)]. For Fe₃O₄@ χ -Fe₅C₂ octahedra, we observed that three surface unit cells of the Fe₃O₄(111) planes (0.48 nm) in the core exhibit good alignment with seven surface unit cells of χ -Fe₅C₂(112) planes (0.21 nm) in the shell (Fig. R1b). Such periodicity leads to a commensurate epitaxial relationship with a low mismatch value of 2.06% according to equation 2. Therefore, the domain match allows the conformal growth for the (112) facet of χ -Fe₅C₂ over the (111) facet of Fe₃O₄.

$$f = 2 \times |7 \times d_{\text{shell}} - 3 \times d_{\text{core}}| / (7 \times d_{\text{shell}} + 3 \times d_{\text{core}}) \quad (2)$$

Figure R1. Lattice matching and domain matching. (a) Lattice matching illustration of Fe₃O₄@ χ -Fe₅C₂ nanocubes. (b) Domain matching illustration of Fe₃O₄@ χ -Fe₅C₂ octahedra.

We have added relevant discussion in the revised manuscript (p. 4, lines 25-30, p. 5, lines 3-4, p. 6, lines 26-31, p. 7, lines 1-12 and 26-28, refs. 33, 35-37, highlighted in yellow color).

“(2) The catalyst structure before and after the reaction should be compared in detail. Does thickness of the layer change?”

As suggested, we have characterized $\text{Fe}_3\text{O}_4@\chi\text{-Fe}_5\text{C}_2$ nanocubes and octahedra after the reaction in detail. The cubic morphology was preserved after 100 h on stream (Fig. R2, a and b). We have measured the thickness of the shell layer for $\text{Fe}_3\text{O}_4@\chi\text{-Fe}_5\text{C}_2$ nanocubes after the reaction. The thickness of the shell increased from 2.0 nm to 2.6 nm after the reaction (Fig. 1c and Fig. R2c). The lattice parameter of the Fe_3O_4 inner core was measured as 0.21 nm, which was indexed as the (400) facet of Fe_3O_4 (Fig. R2d). The lattice parameter of the $\chi\text{-Fe}_5\text{C}_2$ shell was 0.22 nm, which was assigned to the (202) facet of $\chi\text{-Fe}_5\text{C}_2$ (Fig. R2e). The exposed $\chi\text{-Fe}_5\text{C}_2$ facets of $\text{Fe}_3\text{O}_4@\chi\text{-Fe}_5\text{C}_2$ nanocubes were preserved after 100 h on stream. The EELS image implied that the core region mainly comprised Fe and O elements while the shell region contained Fe and C elements (Fig. R2f). We have also conducted Mössbauer spectroscopy to characterize the compositions of the iron phase in the used $\text{Fe}_3\text{O}_4@\chi\text{-Fe}_5\text{C}_2$ nanocubes after 100 h on stream (Fig. R3a). The content of $\chi\text{-Fe}_5\text{C}_2$ increased from 33.2% to 39.8% after the reaction (Table. R1).

With respect to $\text{Fe}_3\text{O}_4@\chi\text{-Fe}_5\text{C}_2$ octahedra, the inter core collapsed after the reaction (Fig. R4, a and b). The thickness of the shell increased from 1.7 nm to 3.5 nm after the reaction (Fig. 3c and Fig. R4c). The lattice parameters of the inner core and the outer shell were measured as 0.48 nm and 0.21 nm, which were assigned to the (111) facet of Fe_3O_4 and (112) facet of $\chi\text{-Fe}_5\text{C}_2$, respectively (Fig. R4, d and e). The exposed $\chi\text{-Fe}_5\text{C}_2$ facet of $\text{Fe}_3\text{O}_4@\chi\text{-Fe}_5\text{C}_2$ octahedra was preserved after the reaction. The distribution of Fe_3O_4 at the core and $\chi\text{-Fe}_5\text{C}_2$ at the shell was supported by the EELS image (Fig. R4f). The content of $\chi\text{-Fe}_5\text{C}_2$ in $\text{Fe}_3\text{O}_4@\chi\text{-Fe}_5\text{C}_2$ octahedra increased from 29.5% to 40.4% after 100 h on stream (Fig. R3b and Table. R1). Therefore, the thickness of the shell layer for both $\text{Fe}_3\text{O}_4@\chi\text{-Fe}_5\text{C}_2$ nanocubes and octahedra increased after the reaction.

Figure R2. Structural characterizations of $\text{Fe}_3\text{O}_4@\chi\text{-Fe}_5\text{C}_2$ nanocubes after 100 h on stream. (a) TEM image of $\text{Fe}_3\text{O}_4@\chi\text{-Fe}_5\text{C}_2$ nanocube/SiC. (b) HAADF-STEM image of an individual $\text{Fe}_3\text{O}_4@\chi\text{-Fe}_5\text{C}_2$ nanocube. (c) Magnified HAADF-STEM image of the region marked by the corresponding boxes in panel b. (d) Intensity profile recorded from the area indicated by the rectangular box in panel c. (e) Intensity profile recorded from the area indicated by the rectangular box in panel c. (f) EELS spectra of a $\text{Fe}_3\text{O}_4@\chi\text{-Fe}_5\text{C}_2$ nanocube in panel c.

Figure R3. Mössbauer spectra characterizations. (a) Mössbauer spectra of $\text{Fe}_3\text{O}_4@ \chi\text{-Fe}_5\text{C}_2$ nanocubes after 100 h on stream. (b) Mössbauer spectra of $\text{Fe}_3\text{O}_4@ \chi\text{-Fe}_5\text{C}_2$ octahedra after 100 h on stream.

Table R1 | Mössbauer parameters of $\text{Fe}_3\text{O}_4@ \chi\text{-Fe}_5\text{C}_2$ nanocubes and $\text{Fe}_3\text{O}_4@ \chi\text{-Fe}_5\text{C}_2$ octahedra after 100 h on stream. The isomer shift (IS), quadrupole splitting (QS), hyperfine field, and spectral contribution are given.

Samples	Phase ascription	Mössbaure parameters			
		IS (mm/s)	QS (mm/s)	Hyperfine field (T)	Spectral contribution
$\text{Fe}_3\text{O}_4@ \chi\text{-Fe}_5\text{C}_2$ nanocubes after reaction	Fe_3O_4 (A)	0.30	-0.01	48.9	40.7%
	Fe_3O_4 (B)	0.65	0.01	45.5	15.4%
	$\chi\text{-Fe}_5\text{C}_2$ (A)	0.22	0.11	21.8	12.9%
	$\chi\text{-Fe}_5\text{C}_2$ (B)	0.17	0.06	18.6	17.0%
	$\chi\text{-Fe}_5\text{C}_2$ (C)	0.20	0.02	11.1	9.9%
	Fe(II)/Fe(III)	0.34	1.07	-	4.1%
$\text{Fe}_3\text{O}_4@ \chi\text{-Fe}_5\text{C}_2$ octahedra after reaction	Fe_3O_4 (A)	0.33	-0.05	48.5	38.7%
	Fe_3O_4 (B)	0.78	-0.09	45.4	16.6%
	$\chi\text{-Fe}_5\text{C}_2$ (A)	0.27	0.11	21.6	14.5%
	$\chi\text{-Fe}_5\text{C}_2$ (B)	0.22	0.04	18.3	15.5%
	$\chi\text{-Fe}_5\text{C}_2$ (C)	0.28	0.06	11.2	10.4%
	Fe(II)/Fe(III)	0.39	1.04	-	4.3%

Figure R4. Structural characterizations of $\text{Fe}_3\text{O}_4@ \chi\text{-Fe}_5\text{C}_2$ octahedra after 100 h on stream. (a) TEM image of $\text{Fe}_3\text{O}_4@ \chi\text{-Fe}_5\text{C}_2$ octahedra/SiC. (b) HAADF-STEM image of an individual $\text{Fe}_3\text{O}_4@ \chi\text{-Fe}_5\text{C}_2$ octahedron. (c) HAADF-STEM image of another $\text{Fe}_3\text{O}_4@ \chi\text{-Fe}_5\text{C}_2$ octahedron. (c) Magnified HAADF-STEM image of the region marked by the corresponding boxes in panel b. (d) Intensity profile recorded from the area indicated by the rectangular box in panel d. (e) Intensity profile recorded from the area indicated by the rectangular box in panel d. (f) EELS spectra of a $\text{Fe}_3\text{O}_4@ \chi\text{-Fe}_5\text{C}_2$ octahedron in panel d.

We have added relevant discussion in the revised manuscript (p. 10, lines 9-20 and 25-31, p. 11, lines 1-3, Supplementary Figs. 17-19 and Supplementary Table 4, highlighted in yellow color).

“(3) Is the surface layer (FeCx) porous?”

Thanks for raising this issue. To investigate the textural properties, we have carried out N_2 physisorption characterizations (Fig. R5, a and b). The pore-diameter distributions were analyzed using the Barrett-Joyner-Halenda (BJH) method. As shown in Figure R5, c and d), the surface layer of both $\text{Fe}_3\text{O}_4@ \chi\text{-Fe}_5\text{C}_2$ nanocubes and octahedra were not porous. The absence of pores at the surface layer was also confirmed by HAADF-STEM image (Fig. R5, e and f). We have also measured the textural properties of spent $\text{Fe}_3\text{O}_4@ \chi\text{-Fe}_5\text{C}_2$ nanocubes and octahedra after 100 h on stream (Fig. R6, a and b). The surface layer of spent $\text{Fe}_3\text{O}_4@ \chi\text{-Fe}_5\text{C}_2$ nanocubes remained nonporous after 100 h on stream (Fig. R6, c and e). In contrast, spent $\text{Fe}_3\text{O}_4@ \chi\text{-Fe}_5\text{C}_2$ octahedra contained mesopores as revealed by the pore-diameter distribution and HAADF-STEM image (Fig. R6, d and f). The average mesopore diameter was determined as 17.9 nm by the BJH method (Fig. R6d).

Figure R5. Textural properties of $\text{Fe}_3\text{O}_4@ \chi\text{-Fe}_5\text{C}_2$ nanocubes and octahedra before reaction. (a, b) Nitrogen adsorption and desorption isotherm of (a) $\text{Fe}_3\text{O}_4@ \chi\text{-Fe}_5\text{C}_2$ nanocubes and (b) $\text{Fe}_3\text{O}_4@ \chi\text{-Fe}_5\text{C}_2$ octahedra. (c, d) Pore-size distributions of (c) $\text{Fe}_3\text{O}_4@ \chi\text{-Fe}_5\text{C}_2$ nanocubes and (d) $\text{Fe}_3\text{O}_4@ \chi\text{-Fe}_5\text{C}_2$ octahedra derived from the nitrogen adsorption-desorption isotherms by the BJH method. (e, f) HAADF-STEM images of an individual (e) $\text{Fe}_3\text{O}_4@ \chi\text{-Fe}_5\text{C}_2$ nanocubes and (f) $\text{Fe}_3\text{O}_4@ \chi\text{-Fe}_5\text{C}_2$ octahedra.

Figure R6. Textural properties of $\text{Fe}_3\text{O}_4@ \chi\text{-Fe}_5\text{C}_2$ nanocubes and octahedra after 100 h on stream. (a, b) Nitrogen adsorption and desorption isotherm of (a) $\text{Fe}_3\text{O}_4@ \chi\text{-Fe}_5\text{C}_2$ nanocubes and (b) $\text{Fe}_3\text{O}_4@ \chi\text{-Fe}_5\text{C}_2$ octahedra after 100 h on stream. (c, d) Pore-size distributions of (c) $\text{Fe}_3\text{O}_4@ \chi\text{-Fe}_5\text{C}_2$ nanocubes and (d) $\text{Fe}_3\text{O}_4@ \chi\text{-Fe}_5\text{C}_2$ octahedra after 100 h on stream derived from the nitrogen adsorption-desorption isotherms by the BJH method. (e, f) HAADF-STM images of an individual (e) $\text{Fe}_3\text{O}_4@ \chi\text{-Fe}_5\text{C}_2$ nanocubes and (f) $\text{Fe}_3\text{O}_4@ \chi\text{-Fe}_5\text{C}_2$ octahedra after 100 h on stream.

We have added relevant discussion in the revised manuscript (p. 11, lines 4-14, Supplementary Figs. 4 and 20, highlighted in yellow color).

“(4) TOFs of nanocubes and octahedra are not the same?”

After being reminded, we have calculated the TOF numbers based on the moles of CO converted per mole of surface Fe atoms per hour (equation 3).

$$\text{TOF} = \text{CO conversion} \times \text{moles of CO in syngas} \times \text{gas-flow rate} \div \text{moles of surface Fe atoms} \quad (3)$$

The moles of Fe atoms on the surface of $\text{Fe}_3\text{O}_4@ \chi\text{-Fe}_5\text{C}_2$ nanocrystals were determined by CO pulse chemisorption measurement. CO pulse chemisorption measurements were performed using a Micromeritics Autochem 2920 chemisorption analyzer with an active loop volume of 0.1 mL. In a typical measurement, 100 mg of $\text{Fe}_3\text{O}_4@ \chi\text{-Fe}_5\text{C}_2$ nanocrystals/SiC were packed into a reactor with a quartz tube. Prior to the test, the samples were cleaned in He with a gas-flow rate of 100 mL min^{-1} at $270 \text{ }^\circ\text{C}$ for 5 h. After cooling down to $50 \text{ }^\circ\text{C}$ under He flow, CO/He pulses (10 vol% CO and 90 vol% He) were injected until adsorption reached saturation. The amount of adsorbed CO was calculated on the difference between the total amount of CO injected and the amount measured at the outlet from the sample. The metal dispersion was calculated by assuming the ratio of CO to surface metal atom as 1:1. The moles of Fe atoms on the surface of $\text{Fe}_3\text{O}_4@ \chi\text{-Fe}_5\text{C}_2$ nanocubes/SiC was $24.1 \text{ } \mu\text{mol g}^{-1}$, higher than that ($19.5 \text{ } \mu\text{mol g}^{-1}$) of $\text{Fe}_3\text{O}_4@ \chi\text{-Fe}_5\text{C}_2$ octahedra/SiC (Fig. R7, a and b). According to equation 3, the TOF number of $\text{Fe}_3\text{O}_4@ \chi\text{-Fe}_5\text{C}_2$ nanocubes/SiC was 645.9 h^{-1} , being 1.7 times as high as that (372.7 h^{-1}) of $\text{Fe}_3\text{O}_4@ \chi\text{-Fe}_5\text{C}_2$ octahedra/SiC.

Figure R7. CO pulse profiles of (a) $\text{Fe}_3\text{O}_4@ \chi\text{-Fe}_5\text{C}_2$ nanocubes/SiC and (b) $\text{Fe}_3\text{O}_4@ \chi\text{-Fe}_5\text{C}_2$ octahedra/SiC.

We have added relevant discussion in the revised manuscript (p. 8, lines 23-29, p. 21, lines 24-29, p. 22, lines 1-5, Supplementary Fig. 12, highlighted in yellow color).

(5) *More info on the products is expected (i.e. olefinic hydrocarbon selectivity).*

As suggested by this reviewer, we have calculated the olefinic hydrocarbon selectivity in the revised manuscript. Under the same reaction condition, the selectivity for $\text{C}_2\text{-C}_4^-$ olefins over $\text{Fe}_3\text{O}_4@ \chi\text{-Fe}_5\text{C}_2$ nanocubes/SiC was 21.6 C%, approaching that (20.6 C%) over $\text{Fe}_3\text{O}_4@ \chi\text{-Fe}_5\text{C}_2$ octahedra/SiC (Table R2). Additionally, the selectivity for $\text{C}_5\text{-C}_{12}^-$ olefins over $\text{Fe}_3\text{O}_4@ \chi\text{-Fe}_5\text{C}_2$ nanocubes/SiC was 17.9 C%, higher than that (11.3 C%) over $\text{Fe}_3\text{O}_4@ \chi\text{-Fe}_5\text{C}_2$ octahedra/SiC (Table R2). As for $\chi\text{-Fe}_5\text{C}_2$ nanoparticles, the $\text{C}_2\text{-C}_4^-$ and $\text{C}_5\text{-C}_{12}^-$ selectivities were 29.8 C% and 16.9 C%, respectively (Table R2). As for $\text{Fe}_3\text{O}_4@ \chi\text{-Fe}_5\text{C}_2$ octahedra/SiC at similar conversion levels of cubic counterpart, the selectivities for $\text{C}_2\text{-C}_4^-$ and $\text{C}_5\text{-C}_{12}^-$ were 18.3 C% and 16.0 C%, respectively.

Table R2. Catalytic properties of $\text{Fe}_3\text{O}_4@ \chi\text{-Fe}_5\text{C}_2$ nanocubes/SiC and octahedra/SiC towards FTS.

Catalysts	Conversion (%)	CO ₂ selectivity (%)	Product selectivity (C%, CO ₂ -free)						Carbon balance (%)
			CH ₄	C ₂ -C ₄ ⁼	C ₂ -C ₄ ^o	C ₅ -C ₁₂ ⁼	C ₅ -C ₁₂ ^o	C ₁₃₊	
Fe ₃ O ₄ @ χ -Fe ₅ C ₂ nanocubes/SiC ^a	45.4	18.7	14.2	21.6	19.4	17.9	24.3	2.6	98.7
Fe ₃ O ₄ @ χ -Fe ₅ C ₂ octahedra/SiC ^a	21.2	12.8	19.3	20.6	32.5	11.3	14.5	1.8	96.5
χ -Fe ₅ C ₂ nanoparticles/SiC ^a	23.6	10.5	15.5	29.8	18.4	16.9	18.0	1.4	96.8
Fe ₃ O ₄ @ χ -Fe ₅ C ₂ octahedra/SiC ^b	42.6	16.3	20.1	18.3	22.0	16.0	21.5	2.1	97.6

^a refers to the conditions of 20 bar, syngas (64 vol% H₂, 32 vol% CO, and 4 vol% Ar), 2400 mL h⁻¹ g_{cat}⁻¹, and 270 °C.

^b refers to the conditions of 20 bar, syngas (64 vol% H₂, 32 vol% CO, and 4 vol% Ar), 800 mL h⁻¹ g_{cat}⁻¹, and 270 °C.

⁼ refers to olefins.

^o refers to paraffins.

We have added relevant discussion in the revised manuscript (p. 9, lines 15-19, p. 9, lines 21-22, p. 10, lines 2-3, Supplementary Table 3, highlighted in yellow color).

“(6) Some sentences are need to be improved. For example, “0.113g of CTAB” should be “CTAB of 0.113g”.”

As suggested, we have improved the sentences in the revised manuscript.

We have added relevant discussion in the revised manuscript (p.19, line 11, p.19, lines 17-19, highlighted in yellow color).

Reviewer #2:

“The authors successfully fabricated two core-shell catalysts with {202} and {112} facets of χ -Fe₅C₂ as the outer shell through the conformal reconstruction of Fe₃O₄ nanocubes and octahedra, as the inner cores. The sensitivity of the facet of iron carbides to performance were explored. The different types of CO dissociation led to different FTS activity. Although some interesting results have been got, there are still some problems need to be addressed.”

We appreciate the reviewer’s comments regarding the structural characterizations. As suggested by this reviewer, we have conducted additional experiments and DFT calculations in the revised manuscript. The details are listed in the following responses.

“1. First and most importantly, as we know, the carbon permeation, diffusion and carburization occur. Therefore, the Fe-based catalysts undergo a long activation period

before the structure is stable. In this work, the authors correlated the activity to the different facets. But no solid evidence confirms that the stable structure has been achieved. The carbon balance data were not provided. So, we can confirm that the different CO conversions are assigned to different activity or continuous carburization or structure evolution. And the CO/H₂ ratio during activation and reaction is different. Even the TEM images for the used catalysts were provided, the Fe₃O₄@ χ -Fe₅C₂ octahedra collapsed, and the nanocrystals seem to be more stable. But is the thickness of the carbide shell the same? Is the exposed facet maintained after reaction? And also, the compositions of iron phase of the used catalysts are missed. This is very critical for the main conclusion of this work. Because it is very common that the “apparent activity” keeps a long time, but the structure of carbides changes a lot.”

As suggested by this reviewer, we calculated the carbon balance value (Table R1). For catalytic tests, the products were detected via an offline method. There is an ice trap to separate liquid and gas products. The gaseous products were monitored by online gas chromatographs. The liquid products were analyzed using an offline chromatograph. The carbon balance value was calculated via equation 1,

$$\text{Carbon balance} = \frac{C_n H_m + CO_2}{CO_{\text{inlet}} - CO_{\text{outlet}}} \times 100\% \quad (1)$$

where C_nH_m and CO₂ represent the moles of the produced hydrocarbons and CO₂, respectively. CO_{inlet} and CO_{outlet} are moles of CO at the inlet and outlet, respectively.

The carbon balance value of Fe₃O₄@ χ -Fe₅C₂ nanocubes/SiC was 98.7%, similar to that (96.5%) of Fe₃O₄@ χ -Fe₅C₂ octahedra/SiC (Table R1). The high carbon balance value indicated that the different CO conversions were not caused by continuous carburization.

Table R1. Catalytic properties of Fe₃O₄@ χ -Fe₅C₂ nanocubes/SiC and octahedra/SiC towards FTS.

Catalysts	Conversion (%)	CO ₂ selectivity (%)	Product selectivity (C%, CO ₂ -free)						Carbon balance (%)
			CH ₄	C ₂ -C ₄ ⁼	C ₂ -C ₄ ^o	C ₅ -C ₁₂ ⁼	C ₅ -C ₁₂ ^o	C ₁₃₊	
Fe ₃ O ₄ @ χ -Fe ₅ C ₂ nanocubes/SiC ^a	45.4	18.7	14.2	21.6	19.4	17.9	24.3	2.6	98.7
Fe ₃ O ₄ @ χ -Fe ₅ C ₂ octahedra/SiC ^a	21.2	12.8	19.3	20.6	32.5	11.3	14.5	1.8	96.5
χ -Fe ₅ C ₂ nanoparticles/SiC ^a	23.6	10.5	15.5	29.8	18.4	16.9	18.0	1.4	96.8
Fe ₃ O ₄ @ χ -Fe ₅ C ₂ octahedra/SiC ^b	42.6	16.3	20.1	18.3	22.0	16.0	21.5	2.1	97.6

^a refers to the conditions of 20 bar, syngas (64 vol% H₂, 32 vol% CO, and 4 vol% Ar), 2400 mL h⁻¹ g_{cat}⁻¹, and 270 °C.

^b refers to the conditions of 20 bar, syngas (64 vol% H₂, 32 vol% CO, and 4 vol% Ar), 800 mL h⁻¹ g_{cat}⁻¹, and 270 °C.

⁼ refers to olefins.

^o refers to paraffins.

We characterized Fe₃O₄@ χ -Fe₅C₂ nanocubes and octahedra in detail after the reaction. The cubic morphology was preserved after 100 h on stream (Fig. R1, a and b). We counted the thickness of the shell layer for Fe₃O₄@ χ -Fe₅C₂ nanocubes after the reaction. The thickness of the shell was increased from 2.0 nm to 2.6 nm before and after the reaction (Fig. 1c and Fig. R1c). The lattice parameter of the Fe₃O₄ inner core was measured as 0.21 nm, which was indexed as the (400) facet of Fe₃O₄ (Fig. R1d). The lattice parameter of the χ -Fe₅C shell was 0.22 nm, which was assigned to the (202) facet of χ -Fe₅C₂ (Fig. R1e). The exposed χ -Fe₅C facet of Fe₃O₄@ χ -Fe₅C₂ nanocubes maintained after 100 h on stream. The electron energy loss spectroscopy (EELS) image implied that the core region mainly comprised Fe and O elements while the shell region contained Fe and C elements (Fig. R1f). We also conducted Mössbauer spectroscopy to characterize the compositions of the iron phase in the used Fe₃O₄@ χ -Fe₅C₂ nanocubes after 100 h on stream (Fig. R2a). The content of χ -Fe₅C₂ increased from 33.2% to 39.8% before and after the reaction (Table. R2). With respect to Fe₃O₄@ χ -Fe₅C₂ octahedra, the inter core collapsed after the reaction (Fig. R3, a and b). The thickness of the shell increased from 1.7 nm to 3.5 nm before and after the reaction (Fig. 3c and Fig. R3c). The lattice parameters of the inner core and the outer shell were measured as 0.48 nm and 0.21 nm, which were assigned to the (111) facet of Fe₃O₄ and (112) facet of χ -Fe₅C₂, respectively (Fig. R3, d and e). The exposed χ -Fe₅C facet of Fe₃O₄@ χ -Fe₅C₂ octahedra preserved after the reaction. The distribution of Fe₃O₄ at the core and χ -Fe₅C₂ at the shell was supported by the EELS image (Fig. R3f). The content of χ -Fe₅C₂ in Fe₃O₄@ χ -Fe₅C₂ octahedra increased from 29.5% to 40.4% before and after 100 h on stream (Fig. R2b and Table. R2). The exposed χ -Fe₅C facet maintained for both Fe₃O₄@ χ -Fe₅C₂ nanocubes and octahedra, while the thickness of the shell layer increased after the reaction. Moreover, The CO conversion of Fe₃O₄@ χ -Fe₅C₂ nanocubes/SiC was 43.1%, much higher than that (20.1%) of Fe₃O₄@ χ -Fe₅C₂ octahedra/SiC at the initial reaction (Fig. R4). Therefore, we assigned different CO conversions to the intrinsic activity of exposed facets of χ -Fe₅C₂.

Figure R1. Structural characterizations of Fe₃O₄@ χ -Fe₅C₂ nanocubes after 100 h on stream. (a) TEM image of Fe₃O₄@ χ -Fe₅C₂ nanocube/SiC. (b) HAADF-STEM image of an individual Fe₃O₄@ χ -Fe₅C₂ nanocube. (c) Magnified HAADF-STEM image of the region marked by the corresponding boxes in panel b. (d) Intensity profile recorded from the area indicated by the rectangular box in panel c. (e) Intensity profile recorded from the area indicated by the rectangular box in panel c. (f) EELS spectra of a Fe₃O₄@ χ -Fe₅C₂ nanocube in panel c.

Figure R2. Mössbauer spectra characterizations. (a) Mössbauer spectra of Fe₃O₄@ χ -Fe₅C₂ nanocubes after 100 h on stream. (b) Mössbauer spectra of Fe₃O₄@ χ -Fe₅C₂ octahedra after 100 h on stream.

Table R2 | Mössbauer parameters of Fe₃O₄@ χ -Fe₅C₂ nanocubes and Fe₃O₄@ χ -Fe₅C₂ octahedra after 100 h on stream. The isomer shift (IS), quadrupole splitting (QS), hyperfine field, and spectral contribution are given.

Samples	Phase ascription	Mössbauer parameters			
		IS (mm/s)	QS (mm/s)	Hyperfine field (T)	Spectral contribution
Fe ₃ O ₄ @ χ -Fe ₅ C ₂ nanocubes after reaction	Fe ₃ O ₄ (A)	0.30	-0.01	48.9	40.7%
	Fe ₃ O ₄ (B)	0.65	0.01	45.5	15.4%
	χ -Fe ₅ C ₂ (A)	0.22	0.11	21.8	12.9%
	χ -Fe ₅ C ₂ (B)	0.17	0.06	18.6	17.0%
	χ -Fe ₅ C ₂ (C)	0.20	0.02	11.1	9.9%
	Fe(II)/Fe(III)	0.34	1.07	-	4.1%
Fe ₃ O ₄ @ χ -Fe ₅ C ₂ octahedra after reaction	Fe ₃ O ₄ (A)	0.33	-0.05	48.5	38.7%
	Fe ₃ O ₄ (B)	0.78	-0.09	45.4	16.6%
	χ -Fe ₅ C ₂ (A)	0.27	0.11	21.6	14.5%

χ -Fe ₅ C ₂ (B)	0.22	0.04	18.3	15.5%
χ -Fe ₅ C ₂ (C)	0.28	0.06	11.2	10.4%
Fe(II)/Fe(III)	0.39	1.04	-	4.3%

Figure R3. Structural characterizations of Fe₃O₄@ χ -Fe₅C₂ octahedra after 100 h on stream. (a) TEM image of Fe₃O₄@ χ -Fe₅C₂ octahedra/SiC. (b) HAADF-STEM image of an individual Fe₃O₄@ χ -Fe₅C₂ octahedron. (c) HAADF-STEM image of another Fe₃O₄@ χ -Fe₅C₂ octahedron. (c) Magnified HAADF-STEM image of the region marked by the corresponding boxes in panel b. (d) Intensity profile recorded from the area indicated by the rectangular box in panel d. (e) Intensity profile recorded from the area indicated by the rectangular box in panel d. (f) EELS spectra of a Fe₃O₄@ χ -Fe₅C₂ octahedron in panel d.

Figure R4. Stability tests of Fe₃O₄@ χ -Fe₅C₂ nanocubes/SiC and octahedra/SiC. The reaction was conducted under 20 bar of syngas (CO:H₂ = 1:2, 2400 mL h⁻¹ g_{cat}⁻¹) at 270 °C.

We have added relevant discussion in the revised manuscript (p. 8, lines 29-30, p. 9, lines 1-2, p. 10, lines 9-20 and 25-31, p. 11, lines 1-3, p. 20, lines 15-19, Supplementary Figs. 17-19, Supplementary Tables 3 and 4, highlighted in yellow color).

“2. The authors claimed that the facets are not sensitive for carbon-chain growth. But the CH₄ production is obviously improved. Is there any explanation?”

We apologize for the not rigorous summary of the facet effect on carbon-chain growth. Actually, the probability of chain growth (α) of Fe₃O₄@ χ -Fe₅C₂ nanocubes/SiC was 0.66, higher than that (0.64) of octahedral counterpart at similar conversion levels. Fe₃O₄@ χ -Fe₅C₂ nanocubes/SiC exhibited slightly higher carbon-chain growth ability compared to the octahedral counterpart. As the distribution of hydrocarbon products followed a typical Anderson-Schulz-Flory (ASF) statistics, the CH₄ selectivity of Fe₃O₄@ χ -Fe₅C₂ nanocubes/SiC was 14.2 C%, lower than that (20.1 C%) of Fe₃O₄@ χ -Fe₅C₂ octahedra/SiC.

We explored the adsorption of CO and H₂ by conducting pulse chemisorption measurements to rationalize why Fe₃O₄@ χ -Fe₅C₂ nanocubes/SiC exhibited better carbon-chain growth ability. The amount of adsorbed gas was calculated on the difference between the total amount of gas injected and the amount measured at the outlet from the sample. The amount of adsorbed CO over Fe₃O₄@ χ -Fe₅C₂ nanocubes/SiC was 24.1 $\mu\text{mol g}^{-1}$, higher than that (19.5 $\mu\text{mol g}^{-1}$) of Fe₃O₄@ χ -Fe₅C₂ octahedra (Fig. R5, a and b). While the amount of adsorbed H₂ over Fe₃O₄@ χ -Fe₅C₂ nanocubes/SiC was 7.3 $\mu\text{mol g}^{-1}$, lower than that (11.0 $\mu\text{mol g}^{-1}$) of octahedral counterpart (Fig. R6, a and b). The results indicated that Fe₃O₄@ χ -Fe₅C₂ nanocubes/SiC improved the CO adsorption and suppressed the H₂ adsorption compared to the octahedral counterpart. Thus, the surface CO/H₂ ratio of Fe₃O₄@ χ -Fe₅C₂ nanocubes/SiC was 3.3, higher than that (1.8) of the octahedral counterpart (Fig. R7). With the increase of the surface CO/H₂ ratio, the CH₄ production of Fe₃O₄@ χ -Fe₅C₂ nanocubes/SiC was suppressed and hydrocarbon products shifted to long-chain molecules.

Figure R5. CO pulse profiles of (a) Fe₃O₄@ χ -Fe₅C₂ nanocubes/SiC and (b) Fe₃O₄@ χ -Fe₅C₂ octahedra/SiC.

Figure R6. H₂ pulse profiles of (a) Fe₃O₄@χ-Fe₅C₂ nanocubes/SiC and (b) Fe₃O₄@χ-Fe₅C₂ octahedra/SiC.

Figure R7. The amount of adsorbed CO and H₂ over Fe₃O₄@χ-Fe₅C₂ nanocubes/SiC and Fe₃O₄@χ-Fe₅C₂ octahedra/SiC.

We have also conducted density functional theory (DFT) calculations to analyze the facet effect in CH₄ production. We used χ-Fe₅C₂(202) and χ-Fe₅C₂(112) surfaces to simulate Fe₃O₄@χ-Fe₅C₂ nanocubes and octahedra, respectively (Figs. R8 and R9). As shown in Fig. R10, the energy barrier of the CH₂*+H* step over χ-Fe₅C₂(202) is 1.03 eV, higher than that (0.70 eV) over χ-Fe₅C₂(112). Thus, compared with Fe₃O₄@χ-Fe₅C₂ nanocubes, the octahedral counterpart benefits the CH₄ production. Nevertheless, the effect of crystal face on selectivity is not as great as that on activity.

Figure R8. Calculated structure models of CH₂*+H* on the χ-Fe₅C₂(202) facet. Models

of (a) $\text{CH}_2^* + \text{H}^*$, (b) TS1, (c) CH_3^* .

Figure R9. Calculated structure models of $\text{CH}_2^* + \text{H}^*$ on the $\chi\text{-Fe}_5\text{C}_2(112)$ facet. Models of (a) $\text{CH}_2^* + \text{H}^*$, (b) TS2, (c) CH_3^* .

Figure R10. Comparison in energy barriers of $\text{CH}_2^* + \text{H}^*$ over $\chi\text{-Fe}_5\text{C}_2(202)$ and $\chi\text{-Fe}_5\text{C}_2(112)$ facets.

We have added relevant discussion in the revised manuscript (p. 13, lines 22-31, p. 14, lines 1-7, p. 21, lines 13-29, p. 22, lines 1-2 and 7-14, Supplementary Figs. 12 and 37-41, highlighted in yellow color).

“3. The pressure of DRIFTS is lower than the reaction evaluation. More important, the H_2 is induced after CO adsorption in DRIFTS experiments. This is very different from the real reaction that the CO and H_2 are co-fed. The hydrogen-assisted CO dissociation usually occurs in the co-existence of CO and H_2 . And even in the results of this work, once CO adsorbed, the dissociation occurred on both octahedra and nanocube. So, how do the authors discriminate the two kinds of CO dissociation?”

We extend our gratitude to the reviewer for his/her insightful comment, which prompted us to further elucidate the CO dissociation mechanisms in our study. Acknowledging the importance of co-feeding CO and H_2 in DRIFTS experiments, we aimed to discern between direct CO dissociation and hydrogen-assisted CO dissociation pathways. To distinguish between these mechanisms, we have conducted *in-situ* DRIFTS experiments under 20 bar of syngas, simulating realistic reaction environments. Prior to testing, thorough cleaning of the

samples in He at 270 °C for 1 h ensured the elimination of any contaminants. Background spectra were acquired under He flow, followed by exposure to a 20-bar syngas mixture (CO/H₂) at 270 °C for 30 min.

Our analysis of the resulting spectra revealed distinctive features indicative of each dissociation pathway. Specifically, the appearance of gaseous CO₂ peaks at 2360 and 2336 cm⁻¹ provided evidence for direct dissociation occurring on both Fe₃O₄@ χ -Fe₅C₂ nanocubes and octahedra (Figure R11, Table R3). Moreover, the presence of CHO* species, as indicated by a peak at 1741 cm⁻¹, was observed solely on the Fe₃O₄@ χ -Fe₅C₂ nanocubes catalyst and absent on the octahedral counterpart (Figure R11, Table R3).

These findings unequivocally demonstrate the discrimination between the two types of CO dissociation. While direct CO dissociation generates CO₂, hydrogen-assisted CO dissociation leads to the formation of oxygen-containing species, specifically CHO*, which was uniquely observed on the nanocube catalyst. This selective occurrence of CHO* species provides compelling evidence for the preferential involvement of the hydrogen-assisted CO dissociation mechanism on the nanocube catalyst.

Figure R11. (a) *In-situ* DRIFTS spectra of Fe₃O₄@ χ -Fe₅C₂ nanocubes and (b) Fe₃O₄@ χ -Fe₅C₂ octahedra after being exposed to 20 bar of syngas for 30 min at 270 °C.

Table R3. Assignment of DRIFTS peaks.

Wavenumber (cm ⁻¹)	Assignment	References
2360, 2336	Gaseous CO ₂	1
2955	Asymmetrical stretching vibration of C-H bonds in CH ₃ *	2, 3
2919	Asymmetrical stretching vibration of C-H bonds in CH ₂ *	4
2849	Symmetrical stretching vibration of C-H bonds in CH ₂ *	5
1741	Stretching vibration of C=O bonds in CHO*	6,7

References

1. Khan, M. U.; Wang, L.; Liu, Z.; Gao, Z.; Wang, S.; Li, H.; Zhang, W.; Wang, M.; Wang, Z.; Ma, C.; Zeng, J. Pt₃Co octapods as superior catalysts of CO₂ hydrogenation. *Angew. Chem. Int. Ed.* **2016**, *55*, 9548-9552.
2. McNab, A. I.; McCue, A. J.; Dionisi, D.; Anderson, J. A. Quantification and

- qualification by *in-situ* FTIR of species formed on supported-cobalt catalysts during the Fischer-Tropsch reaction. *J. Catal.* **2017**, *353*, 286-294.
- Huynh, H. L.; Zhu, J.; Zhang, G.; Shen, Y.; Tucho, W. M.; Ding, Y.; Yu, Z. Promoting effect of Fe on supported Ni catalysts in CO₂ methanation by *in situ* DRIFTS and DFT study. *J. Catal.* **2020**, *392*, 266-277.
 - Han, X.; Zhao, Q.; Gong, H.; Wei, C.; Lv, J.; Wang, Y.; Wang, M.-y.; Huang, S.; Ma, X. Interface-induced phase evolution and spatial distribution of Fe-Based catalysts for Fischer-Tropsch synthesis. *ACS Catal.* **2023**, *13*, 6525-6535.
 - Bahri, S.; Pathak, S.; Upadhyayula, S. Transient HCO/HCOO⁻ species formation during Fischer-Tropsch over an Fe-Co spinel using low Ribblet ratio syngas: a combined operando IR and kinetic study. *Sustain. Energy Fuels* **2023**, *7*, 708-726.
 - Miao, B.; Ma, S. S. K.; Wang, X.; Su, H.; Chan, S. H. Catalysis mechanisms of CO₂ and CO methanation. *Catal. Sci. Technol.* **2016**, *6*, 4048-4058.
 - Hartman, T.; Geitenbeek, R. G.; Whiting, G. T.; Weckhuysen, B. M. Operando monitoring of temperature and active species at the single catalyst particle level. *Nat. Catal.* **2019**, *2*, 986-996.

Moreover, we have also conducted DFT calculations to discriminate the direct dissociation and hydrogen-assisted dissociation of CO in the revised manuscript. Detailed discussion is described in the next response.

We have added relevant discussion in the revised manuscript (p. 12, lines 19-28, p. 21, lines 8-11, Supplementary Fig. 22, Supplementary Table 5, Supplementary refs. 1-7, highlighted in yellow color).

“4. I think if the DFT calculations are given, the conclusion will be more supportive.”

As suggested by this reviewer, we have calculated the energy barriers for the CO direct dissociation route and hydrogen-assisted dissociation path over χ -Fe₅C₂(202) and χ -Fe₅C₂(112) surfaces. For χ -Fe₅C₂(202) surface, the hydrogen-assisted CO dissociation route is the dominating route, since its energy barrier of the rate-limiting step (CO* + H* → HCO*) 1.23 eV is much lower than the direct CO dissociation route with an energy barrier as high as 2.85 eV (Figs. R12-R15). For χ -Fe₅C₂(112) surface, the direct CO dissociation route exhibits a lower energy barrier (1.37 eV) than the hydrogen-assisted CO dissociation route (1.54 eV), implying the direct dissociation as the main route of CO dissociation (Figs. R16-R19). Besides, the energy barrier of the hydrogen-assisted CO dissociation route on the χ -Fe₅C₂(202) surface is lower than that of the direct CO dissociation on the χ -Fe₅C₂(112) surface (Figs. R15 and R19). The DFT conclusion was consistent with the *in-situ* DRIFTS spectra result.

Figure R12. Calculated structure models of the direct CO dissociation route on the χ -Fe₅C₂(202) facet. Models of (a) CO*, (b) TS, and (c) C* + O*.

Figure R13. Calculated structure models of the hydrogen-assisted CO dissociation route on the χ -Fe₅C₂(202) facet. Models of (a) CO* + H*, (b) TS1, (c) HCO*, (d) TS2, and (e) HC* + O*.

Figure R14. Energy barrier of the direct CO dissociation route on the χ -Fe₅C₂(202) facet.

Figure R15. Energy barrier of the hydrogen-assisted CO dissociation route on the χ -Fe₅C₂(202) facet.

Figure R16. Calculated structure models of the direct CO dissociation route on χ -Fe₅C₂(112) facet. Models of (a) CO*, (b) TS, and (c) C* + O*.

Figure R17. Calculated structure models of the hydrogen-assisted CO dissociation route on the χ -Fe₅C₂(112) facet. Models of (a) CO* + H*, (b) TS1, (c) HCO*, (d) TS2, and (e) HC* + O*.

Figure R18. Energy barrier of the direct CO dissociation route on the χ -Fe₅C₂(112) facet.

Figure R19. Energy barrier of the hydrogen-assisted CO dissociation route on the χ -Fe₅C₂(112) facet.

We have added relevant discussion in the revised manuscript (p. 13, lines 11-21, p. 21, lines 13-22, Supplementary Figs. 29-35, highlighted in yellow color).

Reviewer #3:

“In this work, the authors constructed Fe₃O₄@Fe₅C₂ core-shell with different facets which exhibited comparable activity and catalytic mechanism. However, the reconstruction of Fe₅C₂ during Fischer-Tropsch reaction has been systematically investigated in early reported works. (J. Phys. Chem. C 2017, 121, 9, 5154-5160; ACS Catal. 2017, 7, 9, 5661-5667) The synthesis of Fe₃O₄@Fe₅C₂ core-shell for Fischer-Tropsch synthesis has also been reported. (ACS Catal. 2016, 6, 6, 3610-3618) The catalytic mechanism has also been reported. (Applied Energy 2015, 160, 15, 982-989). Therefore, I think the authors should really differentiate their works in novelty from the literature reports before getting publishing on Nat. Commun..”

We genuinely thank this reviewer for the insightful comment. We apologize for any lack of clarity in articulating the novelty of our work. Allow us to elaborate on the distinctive contributions of our research compared to the referenced literature.

Facet sensitivity in practical iron carbides. While previous studies have extensively explored iron carbides' role in FTS, they primarily focused on synthesis methods and general catalytic performance. In contrast, our work delves into the facet sensitivity of practical iron carbides. By synthesizing Fe₃O₄@ χ -Fe₅C₂ nanocrystals with specifically exposed surfaces, we provide concrete evidence supporting facet sensitivity in FTS reactions, a crucial factor only proposed in theoretical calculations or studies on single-crystal surfaces.

Novel synthesis of uniformly exposed surfaces. While the synthesis of Fe₃O₄@ χ -Fe₅C₂ core-shell nanoparticles has been reported previously, our approach stands out for the preparation of nanocrystals with uniformly exposed χ -Fe₅C₂ facets. Unlike previous methods resulting in irregular spherical particles, our technique utilizes a highly symmetrical Fe₃O₄ template to stabilize uniform facet exposure. This achievement represents a significant advancement in the synthesis of iron carbide catalysts, offering precise control over surface characteristics crucial for catalytic performance.

Insights into catalytic mechanisms. While existing literature has explored the transformation of iron phases and their role in FTS, our study goes beyond by elucidating the CO dissociation pathways on different facets of χ -Fe₅C₂. Through detailed DRIFTS measurements, we identify distinct dissociation routes over Fe₃O₄@ χ -Fe₅C₂ nanocubes and octahedra, shedding light on the intricate relationship between facet types and catalytic behavior. This novel insight enhances our understanding of FTS mechanisms and offers valuable guidance for future catalyst design strategies.

In summary, our work significantly advances the field by providing experimental evidence of facet sensitivity in practical iron carbides for FTS, synthesizing nanocrystals with uniformly exposed surfaces, and elucidating distinct catalytic mechanisms based on facet types. We believe these contributions distinguish our research from previous literature reports and underscore its significance in advancing catalysis science.

The differences between our work and the references mentioned by the reviewer are discussed in detail as follows:

The reference (J. Phys. Chem. C 2017, 121, 9, 5154-5160; ACS Catal. 2017, 7, 9, 5661-5667) highlighted how Fe₅C₂ nanoparticles were evolved from Fe(CO)₅ reagent through *in-situ* observation. These articles primarily investigated the synthesis of pure-phase iron carbides. In contrast, our study elucidated the fabrication of Fe₃O₄@ χ -Fe₅C₂ nanocrystals with surfaces terminated in {202} and {112} facets of χ -Fe₅C₂ shells, achieved by utilizing cubic and octahedral Fe₃O₄ as templates, respectively. Our focus lies in the methodology of constructing χ -Fe₅C₂ with uniformly exposed surfaces from Fe₃O₄ via a conformal reconstruction method. Thus, the research content of the references and our work diverges significantly. In short, these works provided mechanistic understandings of the active phase, whereas our work takes a step further by delving into the active facets.

In the reference (ACS Catal. 2016, 6, 6, 3610-3618), the authors synthesized Fe₃O₄@ χ -Fe₅C₂ nanoparticles by pyrolyzing iron-containing metal-organic frameworks, presenting a novel synthetic route for preparing FTS catalysts with the χ -Fe₅C₂ phase. However, the obtained Fe₃O₄@ χ -Fe₅C₂ nanoparticles were irregular spherical particles. Although core-shell nanoparticles of Fe₃O₄@ χ -Fe₅C₂ have been synthesized, the preparation of uniform χ -Fe₅C₂ facets is rare, to the best of our knowledge. χ -Fe₅C₂ nanocrystals tend to assume spherical shapes under pressures up to tens of atmospheres and temperatures up to several hundred °C to minimize surface energy. To stabilize the uniformly exposed facets of χ -Fe₅C₂, we utilized a highly symmetrical Fe₃O₄ template as the core to support the χ -Fe₅C₂ shell. Hence, one of the novelties of our work lies in the first-time synthesis of χ -Fe₅C₂ nanocrystals with uniformly exposed surfaces.

The reference (Appl. Energy 2015, 160, 15, 982-989) reported the positive role of the transformation of reduced iron phases to iron carbides in promoting the formation of hydrocarbon species. The authors characterized the microstructures at different FTS stages via DRIFTS. In our study, we reported the conformal reconstruction of well-defined Fe₃O₄ nanocrystals to generate χ -Fe₅C₂ with specifically exposed surfaces. Regarding the catalytic mechanism, the reference observed an increase in the amounts of hydrocarbon species absorbed on the catalyst as the temperature rose. In contrast, our work explored CO dissociation pathways for different facets of χ -Fe₅C₂ through DRIFTS measurements. By analyzing the intermediates generated on uniform χ -Fe₅C₂ facets, we distinguished between direct and hydrogen-assisted CO dissociation routes over Fe₃O₄@ χ -Fe₅C₂ nanocubes and

octahedra. Thus, our study provides clarity regarding the different roles and functions of different χ -Fe₅C₂ facets.

We have added relevant discussion in the revised manuscript (p. 3, lines 7-11, refs. 17-20, highlighted in yellow color).

Some specific points:

1. In page 8 line 221, the authors claimed that “the solid octahedra collapsed after the reaction.” However, In Fig. S16, it is easy to find the collapsed nanocubes while the CO conversion was nearly constant. The difference of stability of two samples could not be only contributed to the collapsed morphology.

We sincerely thank this reviewer for his/her constructive suggestions. To investigate other underlying mechanisms for catalyst deactivation, we have employed Raman spectroscopy to analyze the surfaces of Fe₃O₄@ χ -Fe₅C₂ nanocubes and octahedra after 100 h. The presence of peaks at 1330 cm⁻¹ indicated the existence of disordered carbon (D band), while those at 1592 cm⁻¹ signified graphite (G band) (Fig. R1, a and b). Significantly higher intensities of these peaks were observed on the surface of spent Fe₃O₄@ χ -Fe₅C₂ octahedra compared with spent nanocubes, suggesting a greater accumulation of deposited carbon on the octahedral surface (Fig. R1, a and b).

For a more precise comparison, we have conducted thermogravimetric analysis (TGA) under the N₂ atmosphere on both Fe₃O₄@ χ -Fe₅C₂ nanocubes and octahedra after the reaction. The weight loss between 200 and 500 °C was attributed to the removal of long-chain hydrocarbons from the surface, while the weight loss beyond 500 °C was associated with the loss of deposited carbon. In the case of spent Fe₃O₄@ χ -Fe₅C₂ nanocubes, a weight loss of 6.2 wt% was observed (Fig. R1c). Conversely, during TGA testing, spent Fe₃O₄@ χ -Fe₅C₂ octahedra exhibited a total weight loss of 13.5 wt% due to long-chain hydrocarbons and deposited carbon (Fig. R1d). The higher residual weight of long-chain hydrocarbons and deposited carbon on the octahedral structure suggests a more pronounced blockage of active sites compared to nanocubes. Hence, we hypothesize that carbon deposition also contributes to the deactivation of Fe₃O₄@ χ -Fe₅C₂ octahedra.

Figure R1. Raman spectra of $\text{Fe}_3\text{O}_4@ \chi\text{-Fe}_5\text{C}_2$ nanocubes (a) and $\text{Fe}_3\text{O}_4@ \chi\text{-Fe}_5\text{C}_2$ octahedra (b) after 100 h on stream. TGA profiles of $\text{Fe}_3\text{O}_4@ \chi\text{-Fe}_5\text{C}_2$ nanocubes (c) and $\text{Fe}_3\text{O}_4@ \chi\text{-Fe}_5\text{C}_2$ octahedra (d) after 100 h on stream.

We have added relevant discussion in the revised manuscript (p. 11, lines 15-31, p. 12, line 1, p. 22, lines 24-25, Supplementary Fig. 21, highlighted in yellow color).

“2. Schematic diagram for different Fischer-Tropsch mechanisms of $\text{Fe}_3\text{O}_4@ \chi\text{-Fe}_5\text{C}_2$ with different facets should be provided to help the readers to understand the molecular catalytic mechanisms.”

Thanks for this reviewer’s constructive suggestion. As suggested, we have drawn the schematic diagram for different Fischer-Tropsch mechanisms of $\text{Fe}_3\text{O}_4@ \chi\text{-Fe}_5\text{C}_2$ nanocrystals with different facets (Fig. R2). The $\text{Fe}_3\text{O}_4@ \chi\text{-Fe}_5\text{C}_2$ octahedra enabled the direct dissociation of CO, while both direct and hydrogen-assisted CO dissociation routes existed on $\text{Fe}_3\text{O}_4@ \chi\text{-Fe}_5\text{C}_2$ nanocubes.

Figure R2. The schematic diagram for different CO dissociation pathways of $\text{Fe}_3\text{O}_4@ \chi\text{-Fe}_5\text{C}_2$ nanocubes and $\text{Fe}_3\text{O}_4@ \chi\text{-Fe}_5\text{C}_2$ octahedra.

We have added relevant discussion in the revised manuscript (p. 12, lines 25-28, Supplementary Fig. 23, highlighted in yellow color).

REVIEWERS' COMMENTS

Reviewer #1 (Remarks to the Author):

I read the revised manuscript along with the Answer. The authors answered my questions in detail with many additional experimental data. Corresponding revision was also fabricated in the revised manuscript. It is acceptable with its present status.

Reviewer #2 (Remarks to the Author):

Although the authors made many efforts to improve the work, there are still many problems.

1. The solid octahedra collapsed after reaction, but the exposed χ -Fe₅C₂ facet was maintained. I can not understand how the process is.
2. Since the iron carbides are very sensitive to the atmosphere. How did the authors treat the samples after reaction and storage it before characterizations?
3. Some experiments were conducted unreasonably. For example, Supplementary Figure 21, TG were conducted in N₂. It is impossible to remove deposited carbon in N₂ atmosphere.
4. The author used DFT to calculate the energy of the CH₂*+H*, trying to explain the CH₄ selectivity. Why did they use CH₃*+H*? And if the authors would like to explain the selectivity, the C1-C1 should be considered.
5. Figure R6, the authors conducted H₂ chemisorption but only with 6 pulses. It seems that the signal wasn't stable on both two samples. So, the results are not reliable.
6. Table R2, when the GHSV was changed from 2400 to 800 mL h⁻¹ gcat⁻¹, the CO conversion increased from 21.2% to 42.6%. It means longer residence time resulted in lower turnover number, why?

Reviewer #3 (Remarks to the Author):

Because the authors have well addressed the reviewers' comments in the revised paper by performing additional experiments as well as providing rational interpretations, I would like to support the acceptance of this work for publication on the journal in its present content.

Point-by-point response to reviewers' comments
Manuscript ID: NCOMMS-23-62544B
Title: Facet sensitivity of iron carbides in Fischer-Tropsch synthesis

Reviewer #1

"I read the revised manuscript along with the Answer. The authors answered my questions in detail with many additional experimental data. Corresponding revision was also fabricated in the revised manuscript. It is acceptable with its present status."

We sincerely thank this reviewer for his/her careful reading of our manuscript.

Reviewer #2

"Although the authors made many efforts to improve the work, there are still many problems."

We sincerely thank this reviewer's valuable comments on our work. We have clarified the comments raised by this reviewer as follows.

"1. The solid octahedra collapsed after reaction, but the exposed χ -Fe₅C₂ facet was maintained. I can not understand how the process is."

Thanks for raising this issue. The exposed χ -Fe₅C₂ facet of Fe₃O₄@ χ -Fe₅C₂ octahedra was preserved after the reaction, while the structures were porous. The formation of Fe₃O₄@ χ -Fe₅C₂ octahedra with multiple voids can be understood in terms of different diffusion rates of the elements between Fe₃O₄ phase and χ -Fe₅C₂ phase. The mechanism occur via what is known as the nanoscale Kirkendall effect [*Chem. Soc. Rev.* **48**, 1874 (2019); *Appl. Surf. Sci.* **615**, 156269 (2023); *Materials* **15**, 1557 (2022)]. The catalysts were core-shell structure consisting of Fe₃O₄ core and χ -Fe₅C₂ shell at the initial stage of FTS. As the blockage of active sites by long-chain hydrocarbons and non-graphitic carbon, Fe₃O₄@ χ -Fe₅C₂ octahedra were gradually deactivated. Meanwhile, the carbon chemical potential of reaction conditions changed. The dynamic balance of the hydrocarbon production, surface oxidation, and carburization in the syngas environment was broken. This led to the diffusion of Fe atoms between the Fe₃O₄ core and χ -Fe₅C₂ shell. The void formation in Fe₃O₄@ χ -Fe₅C₂ octahedra due to differential diffusion rates of Fe atoms.

We have added the relevant discussion in the revised manuscript (p. 10, lines 25-31, refs 39-41, highlighted in yellow color).

"2. Since the iron carbides are very sensitive to the atmosphere. How did the authors treat the samples after reaction and storage it before characterizations?"

Thanks for raising the question. We used inert gas to protect and store the catalyst after reaction. Specifically, we switched the feed gas to N₂ when the reaction was stopped. The reaction tubes were sealed via valves at both ends after cooling the reactor to room temperature. Afterwards, the reaction tubes were transferred to a N₂-filled glove box. The samples were stored in the glove

box before characterizations. We have added the relevant description in the revised Method section (p. 15, lines 19-24, highlighted in yellow color).

“3. Some experiments were conducted unreasonably. For example, Supplementary Figure 21, TG were conducted in N₂. It is impossible to remove deposited carbon in N₂ atmosphere.”

We sincerely thank this reviewer for his/her valuable comment. The reason that we conducted TGA under N₂ atmosphere was to quantitatively study the blocking degree of the active sites on the χ -Fe₅C₂ surface. We need to heat the sample in an inert atmosphere to ensure that the weight of the catalyst itself does not change. As reported in the literature, the weight loss above 500 °C is ascribed to the removal of the non-graphitic carbon [*Appl. Catal. A-Gen.* **388**, 168-178 (2010)]. In order to describe the type of removed carbon deposit more accurately, we have replaced the deposited carbon in the original manuscript with non-graphitic carbon (p. 12, line 1, highlighted in yellow color).

“4. The author used DFT to calculate the energy of the CH₂+H*, trying to explain the CH₄ selectivity. Why did they use CH₃*+H*? And if the authors would like to explain the selectivity, the C1-C1 should be considered.”*

Thanks for raising this point. We chose to calculate the energy barrier for CH₂*+H* to evaluate methane selectivity because iron-based catalysts generally undergo chain growth by carbide mechanism. The carbide mechanism has been historically associated with CH₂* monomers, since the co-feed of CH₂N₂ or CH₂Cl₂ in syngas was shown to promote chain growth [*ACS Catal.* **9**, 6571-6582 (2019)]. Thus, CH₂ were examined as a potential reaction species to assess the hydrogenation ability of carbon species according to the linear scaling relationship.

As suggested by the reviewer, we have compared the energy barriers of the CH₂*+CH₂* (representing the chain growth) with CH₂*+H* (representing the chain termination) in the revised manuscript. As shown in Figure R1, the energy barrier of the CH₂*+CH₂* step over χ -Fe₅C₂(202) facet is 0.70 eV, lower than that (1.03 eV) of CH₂*+H*. As for χ -Fe₅C₂(112) facet, the energy barrier of the CH₂*+CH₂* step is 0.60 eV, approaching to that (0.7 eV) of CH₂*+H*. Thus, the χ -Fe₅C₂(202) facet is endowed with enhanced chain growth ability than the χ -Fe₅C₂(112) facet.

Figure R1. Calculated structure models and energy barriers of $\text{CH}_2^* + \text{CH}_2^*$. Models of (a) $\text{CH}_2^* + \text{CH}_2^*$, (b) TS, (c) C_2H_4^* on the $\chi\text{-Fe}_5\text{C}_2(202)$ facet. (d) Energy barrier of $\text{CH}_2^* + \text{CH}_2^*$ over $\chi\text{-Fe}_5\text{C}_2(202)$ facet. Models of (e) $\text{CH}_2^* + \text{CH}_2^*$, (b) TS, (c) C_2H_4^* on the $\chi\text{-Fe}_5\text{C}_2(112)$ facet. (d) Energy barrier of $\text{CH}_2^* + \text{CH}_2^*$ over $\chi\text{-Fe}_5\text{C}_2(112)$ facet.

We have added the relevant discussion in the revised manuscript (p. 14, lines 13-15, Supplementary Fig. 42, highlighted in yellow color).

“5. Figure R6, the authors conducted H_2 chemisorption but only with 6 pulses. It seems that the signal wasn't stable on both two samples. So, the results are not reliable.”

We genuinely thank this reviewer for the insightful comment. Actually, we had conducted H_2 chemisorption with 10 pulses but only showed initial 6 pulses in the original manuscript, because we proposed that the adsorption had already reached saturation after 6 pulses. The amount of adsorbed H_2 was calculated on the difference between the total amount of injected H_2 and that at the outlet. When the peak area of desorbed hydrogen remained constant between three pulses, we confirm that the adsorption reaches saturation. Although the height of peaks might seem unstable, the integral area of the last three pulses were kept essentially constant in the original manuscript. When we calculated the amount of adsorbed H_2 by counting 10 pulses, the values were still $7.3 \mu\text{mol g}^{-1}$ for $\text{Fe}_3\text{O}_4@ \chi\text{-Fe}_5\text{C}_2$ nanocubes/SiC and $11.0 \mu\text{mol g}^{-1}$ for $\text{Fe}_3\text{O}_4@ \chi\text{-Fe}_5\text{C}_2$ octahedra/SiC (Fig. R2). For accuracy, we have added the raw data concerning 10 pulses in the revised manuscript (Supplementary Fig. 37).

Figure R2. H₂ pulse profiles of (a) Fe₃O₄@χ-Fe₅C₂ nanocubes/SiC and (b) Fe₃O₄@χ-Fe₅C₂ octahedra/SiC.

To verify the reliability of the results, we have re-conducted H₂-pulse experiments by increasing the concentration of H₂ in the feed gas. The proportion of H₂ in H₂/Ar pulse was increased from 10% to 30%. As shown in Figure R3, the amount of adsorbed H₂ over Fe₃O₄@χ-Fe₅C₂ nanocubes/SiC and Fe₃O₄@χ-Fe₅C₂ octahedra/SiC were calculated as 7.3 μmol g⁻¹ and 11.0 μmol g⁻¹, just equal to the original values (7.3 μmol g⁻¹ and 11.0 μmol g⁻¹). This result implies the reliability of our calculated H₂ adsorption amounts.

Figure R3. H₂ pulse profiles of (a) Fe₃O₄@χ-Fe₅C₂ nanocubes/SiC and (b) Fe₃O₄@χ-Fe₅C₂ octahedra/SiC.

“6. Table R2, when the GHSV was changed from 2400 to 800 mL h⁻¹ g⁻¹, the CO conversion increased from 21.2% to 42.6%. It means longer residence time resulted in lower turnover number, why?”

Thanks for raising this point. I guess that the reviewer might refer to the turnover frequency (TOF) number which reflects the intrinsic activity. Since the intrinsic activity of a catalyst refers exclusively to its ability to facilitate chemical transformations, catalysts must be evaluated at conditions under which rates are not impacted, let alone controlled, by mass and heat transport. A common practice is to operate the reaction in the kinetic control typically with a CO conversion of less than 5%. The conversions of 21.2% and 42.6% largely deviate from the kinetic control interval. As such, we cannot use the current conversion values and GHSV to calculate the TOF number. Based on these values, we can only obtain the apparent mass activity of 5.5 mmol h⁻¹

$\text{g}_{\text{cat}}^{-1}$ at $2400 \text{ mL h}^{-1} \text{ g}_{\text{cat}}^{-1}$ and $3.7 \text{ mmol h}^{-1} \text{ g}_{\text{cat}}^{-1}$ at $800 \text{ mL h}^{-1} \text{ g}_{\text{cat}}^{-1}$. In generaly, the intrinsic activity is the theoretical maximum apparent activity, since the activity can be restrained by the mass and heat transport under a high conversion level. In other words, the lower the conversion, the higher the apparent activity. Thus, the longer residence time resulted in a lower apparent mass activity due to the limitation of mass and heat transport.

Reviewer #3

“Because the authors have well addressed the reviewers' comments in the revised paper by performing additional experiments as well as providing rational interpretations, I would like to support the acceptance of this work for publication on the journal in its present content.”

We genuinely thank the reviewer’s careful reading of our manuscript.